# Assisting Human Decisions in Document Matching

**Joon Sik Kim**  *joonkim@cmu.edu*
*Carnegie Mellon University*

**Valerie Chen**  *valeriechen@cmu.edu*
*Carnegie Mellon University*

**Danish Pruthi**  *danishp@iisc.ac.in*
*Indian Institute of Science, Bangalore*

**Nihar B. Shah**  *nihars@andrew.cmu.edu*
*Carnegie Mellon University*

**Ameet Talwalkar**  *talwalkar@cmu.edu*
*Carnegie Mellon University*

**Reviewed on OpenReview:** *https://openreview.net/forum?id=5rq8iRzHAQ*

## Abstract

Many practical applications, ranging from paper-reviewer assignment in peer review to job-applicant matching for hiring, require human decision makers to identify relevant matches by combining their expertise with predictions from machine learning models. In many such model-assisted document matching tasks, the decision makers have stressed the need for assistive information about the model outputs (or the data) to facilitate their decisions. In this paper, we devise a proxy matching task that allows us to evaluate which kinds of assistive information improve decision makers' performance (in terms of accuracy and time). Through a crowdsourced ($N = 271$ participants) study, we find that providing black-box model explanations reduces users' accuracy on the matching task, contrary to the commonly-held belief that they can be helpful by allowing better understanding of the model. On the other hand, custom methods that are designed to closely attend to some task-specific desiderata are found to be effective in improving user performance. Surprisingly, we also find that the users' perceived utility of assistive information is misaligned with their objective utility (measured through their task performance).

## 1 Introduction

An important application in which human decision makers play a critical role, is document matching, i.e., when a *query document* needs to be matched to one of the many *candidate documents* from a larger pool based on their relevance. Concrete instances of this setup include: *academic peer review*, where meta-reviewers—associate editors in journals (e.g., `https://jmlr.org/tmlr/ae-guide.html`) or area chairs in conferences (Shah, 2022) or program directors conducting proposal reviews (Kerzendorf et al., 2020)—are asked to assign one or more candidate reviewers to submitted papers with relevant expertise based on their previously published work (illustrated in Figure 1, solid arrows); *recruitment*, where recruiters screen through a list of resumes from candidate applicants for an available position at the company (Schumann et al., 2020; Poovizhi et al., 2022); and *plagiarism detection*, where governing members (e.g., ethics board members of a conference, instructors of a course) review submissions to determine the degree of plagiarism (Foltỳnek et al., 2019). Because the pool of candidate documents is typically large and the decision makers have limited time, they first use automated matching models to pre-screen the candidate documents. These matching models typically base their screening on affinity scores, which measure the relevance of each candidate document to

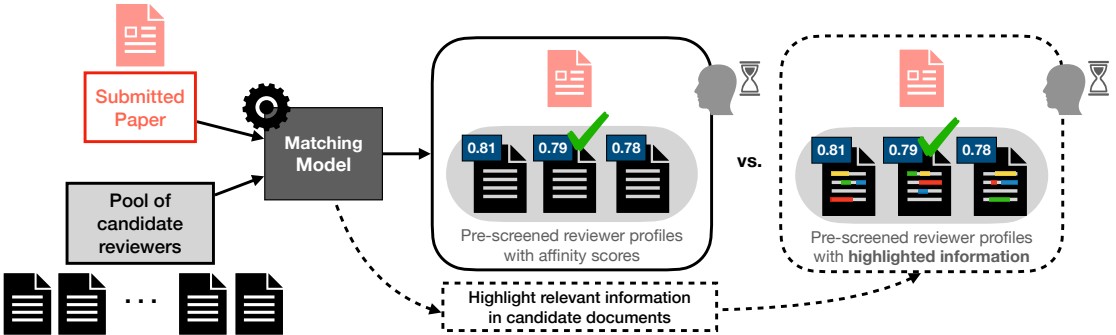

Figure 1: An example document matching application of peer review. For each submitted paper, the matching model pre-screens a list of candidate reviewers via affinity scores (solid arrows). Meta-reviewers, typically under a time constraint, then select the best match to the submitted paper among the pre-screened reviewer (box with a solid line). We study whether providing additional assistive information, namely highlighting potentially relevant information in the candidate documents, can help the meta-reviewers make better decisions (dotted arrows and boxes). We do so by focusing on a proxy matching task on a crowdsourcing platform that is representative of real-world applications not limited to peer review, including recruitment and plagiarism detection which follow the similar setup with different documents and decision makers.

the query document (Alzahrani et al., 2012; Charlin & Zemel, 2013; Cohan et al., 2020; Li et al., 2021). The human decision makers subsequently determine the best-matching document, taking both their expertise and the affinity scores computed by the matching models into account. Such intervention by human decision makers is required for such tasks, as often times either errors made by the models are so consequential that they warrant human oversight, or the overall performance can be considerably improved by incorporating the domain knowledge of human experts.

Despite the growing prevalence of automated matching models and human decision makers working jointly for such practical matching tasks, humans generally find it difficult to completely rely on the models due to a lack of assistive information other than the models' output itself. For instance, in peer review, 20% of the meta-reviewers from past NLP conferences found the affinity scores from the matching model to be *"not very useful or not useful at all"* in a recent survey (Thorn Jakobsen & Rogers, 2022). The survey also reports that the affinity scores rank the least important for the respondents, compared to more tangible and structured information about the candidate reviewers such as whether they have worked on similar tasks, datasets, or methods. Additionally, the survey finds that providing just the affinity scores increases the meta-reviewers' workload as they *"have to identify the information they need from a glance at the reviewers' publication record."* and *"are presented with little structured information about the reviewers."* Similarly, in hiring, the recruiters need to manually evaluate more profiles further down the search result pages due to too generalized matches suggested by the model (Li et al., 2021).

To address the lack of additional assistive information in the document matching setup, we conduct the first evaluation of what additional information can help the human decision makers to find matches *accurately* and *quickly*, compared to when given no additional information (Figure 1, dotted arrows). To do so, we first design a proxy task of summary-article matching that is representative of the general setup so that several methods providing different types of assistive information can be readily tested at scale via crowdsourced users (Section 3.1). The choice of proxy task addresses the logistical difficulty and expenses of directly experimenting with real domain-specific decision makers.

On this proxy task, we explore different classes of methods that have been previously suggested as tools for users to understand model outputs or document content. To standardize the format of assistance, we focus on methods that highlight assistive information within the candidate documents that the decision makers can utilize for matching (Section 3.2):

- SHAP (Lundberg & Lee, 2017), a popular *black-box model explanation* (Doshi-Velez & Kim, 2017; Chen et al., 2022), highlights input tokens in the document that contribute both positively and

negatively to the affinity scores. The utility of SHAP on several concrete downstream tasks remain controversial with conflicting results (Kaur et al., 2020; Jesus et al., 2021; Amarasinghe et al., 2022), and has yet to be evaluated for its effectiveness in document matching.

- BERTSum (Liu & Lapata, 2019), a state-of-the-art *text summarization method*, which highlights key sentences in the candidate documents to help reduce the user's cognitive load for the task.

- Two task-specific methods, that we design ourselves (Section 3.2), to highlight details in the candidate documents relevant to the details in the query (by using sentence and phrase-level similarity measures).

With assistive information provided by these methods as treatments, and a control group provided with just the affinity scores and no additional assistive information, we conduct a pre-registered user study (with 271 participants) on a crowdsourcing platform.[1] The study finds that (Section 4):

- Despite its usage in numerous applications, SHAP decreases the participants' matching performance compared to the control group.

- Contrary to the expectation that summarizing long articles could improve task efficiency, the summaries generated by BERTSum adversely impact the participants. Participants take longer to finish and are less accurate compared to the control group.

- Our task-specific methods, which are tailored to better identify details useful for the task, help the participants to be quicker and more accurate compared to the control group.

- An overwhelming number of participants in *all* treatment groups perceive that the highlighted information is helpful, whereas the quantitative performance (accuracy and time) says otherwise.

The results suggest the benefits of designing task-specific assistive tools over general black-box solutions, and highlight the importance of quantitative evaluation of the methods' utility that is grounded on a specific task over subjective user perceptions (Chen et al., 2022). The code used for the study is available at `https://github.com/wnstlr/document-matching`.

## 2 Related Work

**Prior Evaluation of Assistive Information.** We discuss how our proposed evaluation of different types of assistive information, which include affinity scores, black-box model explanations, and text summaries, differs from how they have been previously evaluated.

Affinity scores, computed by comparing the similarity of representations learned by language models, are commonly used in practice to rank or filter the candidate documents (Mimno & McCallum, 2007; Rodriguez & Bollen, 2008; Charlin & Zemel, 2013; Tran et al., 2017; Wieting et al., 2019; Cohan et al., 2020) Their quality has been evaluated both with or without human decision makers: some may evaluate them based on the user's self-reported confidence score (Mimno & McCallum, 2007), while others may use performance from proxy tasks like document topic classification, where a higher test accuracy of the classification model using the learned representation indicates better ability to reflect more meaningful components in the documents (Cohan et al., 2020). However, the utility of affinity scores for assisting human decision makers for the document matching task is less studied.

While information provided by black-box model explanations have been evaluated for their utility to assist human decision makers in various downstream tasks, the results have been lackluster. On the deception detection task, where users are asked to determine if a given hotel review is fake or not, prior work have shown that only some explanation methods improve a user's task performance (Lai & Tan, 2019; Lai et al., 2020). Arora et al. (2022) further show that none of the off-the-shelf explanations help the users better understand the model's decisions on the task. On more common NLP tasks like sentiment classification

---

[1]Pre-registration document is available here: `https://aspredicted.org/LMM_4K9`

and question-answering, providing explanations to the users decreases the task performance compared to providing nothing when the model's prediction is incorrect (Bansal et al., 2021). For the fraud detection task with domain experts, providing some model explanations showed conflicting effects on improving the performance (Jesus et al., 2021; Amarasinghe et al., 2022). In this work, we expand user evaluations of black-box model explanations to the document matching task and propose alternatives that could be more helpful.

Summaries generated by text summarization models (Lewis et al., 2020; Liu & Lapata, 2019; Shleifer & Rush, 2020; Zhang et al., 2020) are typically either evaluated by quantitative metrics like ROUGE (i.e., statistics based on the amount of overlapping tokens between the generated text and the target text) with respect to the annotated ground-truth summary in a standardized dataset, or by a human's subjective rating of the quality. To the best of our knowledge, the usefulness of these automatically summarized information to the human decision makers in concrete downstream tasks is rarely studied. Even for a few applied works that utilize these methods to practical documents in legal or business domains, the final evaluations do not explore beyond these task-independent metrics (Elnaggar et al., 2018; Bansal et al., 2019; Huang et al., 2020). In this work, we explicitly evaluate whether the generated summaries can help improve the decision makers' task performance in document matching.

**Practical Concerns in Document Matching Applications.** There are a number of real-world document matching applications including peer review, hiring, and plagiarism detection. For each application, we discuss practical issues that have been raised by users that can be mitigated by providing more assistive information about the data and the model.

In scientific peer review, submitted papers need to be matched to appropriate reviewers with proper expertise or experience in the paper's subject area. First, a set of candidate reviewers are identified using an affinity scoring model based on representations learned by language models (Charlin & Zemel, 2013; Mimno & McCallum, 2007; Rodriguez & Bollen, 2008; Tran et al., 2017; Wieting et al., 2019; Cohan et al., 2020). Additional information such as reviewer bids or paper/reviewer subject areas may also be elicited (Shah et al., 2018; Meir et al., 2020; Fiez et al., 2020). Based on this information, meta-reviewers may either be asked to directly assign one or more reviewers to each paper, or to modify the assignment that has been already made as they see appropriate. For example, in the journal Transactions on Machine Learning Research, for any submitted paper the meta-reviewer (action editor) is shown a list of all non-conflicted reviewers sorted according to the affinity scores. The meta-reviewer may also click on any potential reviewer's name to see their website or list of publication. The meta-reviewer is then required to assign three reviewers to the paper based on this information. However, a recent survey of meta-reviewers from past NLP conferences reveal that the affinity scores alone are not as useful, and most respondents prefer to see more tangible and structured information about the reviewers (Thorn Jakobsen & Rogers, 2022).

In hiring, many companies resort to various algorithmic tools to efficiently filter and search for suitable candidates for a given job listing (Fernández & Fernández, 2019; Black & van Esch, 2020; Poovizhi et al., 2022). Many recruiters, while using these tools, express difficulties in reconciling a mismatch between algorithmic results and the recruiter's own assessments. This is mainly attributed to "too generalized and imprecise" relevant matches suggested by the model, which lead to more "manually evaluating more profiles further down the search result pages" increasing the task completion time (Li et al., 2021). Also, the general lack of understanding about the algorithmic assessments makes the recruiters more reluctant to adopt them.

In plagiarism detection, many existing software tools aim to reduce the governing members' workload by providing detailed information about the match, e.g., what specific parts of the query document are identical or similar to parts of the candidate documents. However, their performance in identifying various forms of plagiarism (e.g., ones involving paraphrasing or cross-language references) is still limited (Jiffriya et al., 2021). Also many existing tools lack user-friendly presentation of information that can better assist the task (Foltỳnek et al., 2020). As the governing members need to ultimately assess the proposed evidence by the model to determine the degree of penalty (Foltỳnek et al., 2019), additional assistive information about the match may improve their experience.

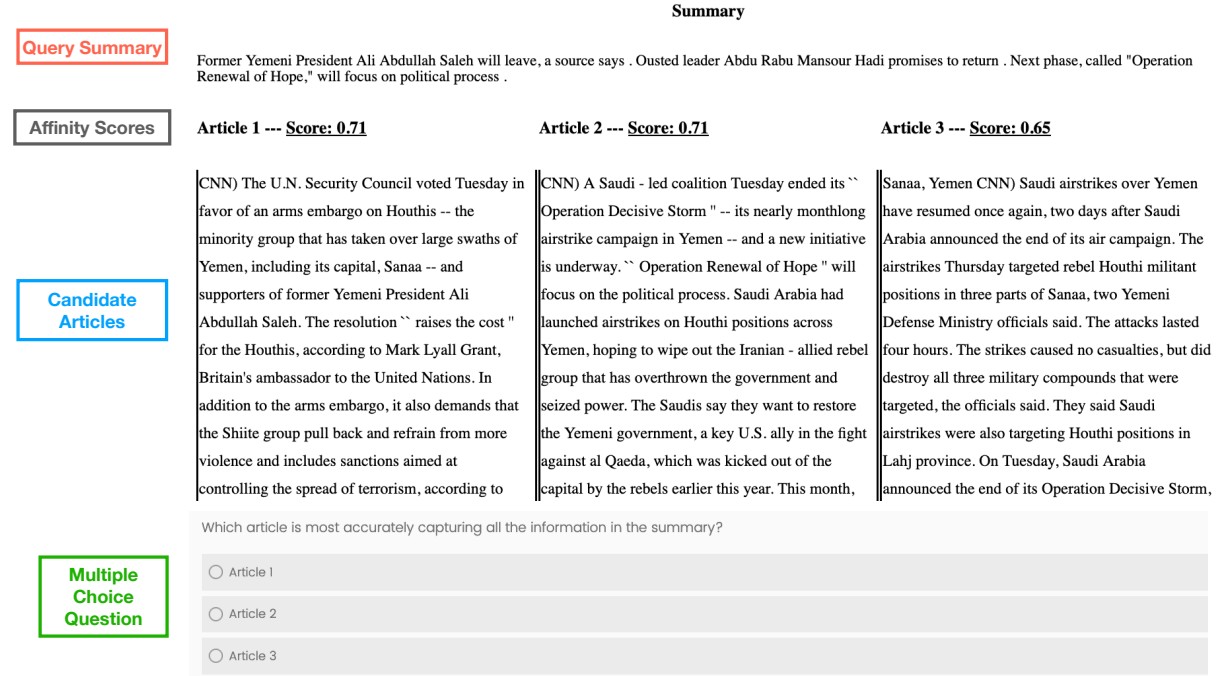

Figure 2: Interface for our summary-article matching task, an instance of the general document matching task. For each question, the participants are provided with the summary, three candidate articles to select from, and affinity scores for each candidate. The articles here are abridged to save space.

## 3 Task Setup and Methods

In Section 3.1, we describe the design of a summary-article matching task, which is an instance of the document matching tasks. We use this task as a proxy for other document matching tasks (e.g., matching reviewers to papers in peer-review), as it is more amenable for crowdsourcing experiments at scale. The summary-article matching task addresses common difficulties encountered when directly experimenting on real-world applications like recruiting real domain-specific decision makers (e.g., meta-reviewers in academia), building on complex systems in practice (e.g., internal systems that govern workflows in academic conferences), and coordinating logistical issues (e.g., longer turnaround for receiving feedback for each paper assignment). Our task may also be useful for early prototyping and validation of different methods. Then in Section 3.2, we present existing and our proposed methods that provide assistive information that we evaluate with human users on the the summary-article matching task.

### 3.1 Instantiating Document Matching

In the general document matching task, a matching model pre-selects a set of candidate documents based on the affinity scores, which capture the relevance between the query document and the candidate document. These affinity scores facilitate filtering candidates from a large pool of documents, but are nevertheless prone to errors. The user therefore goes over the candidate documents with the scores and selects the most relevant candidate document. A practical concern which we would like to address is when the affinity scores from the matching model alone may not provide sufficient information to determine a match quickly and accurately. We outline how we instantiate the summary-article matching task that captures these details.

**Task setup.** We instantiate the general document matching tasks with a *summary-article matching task*. Here, the query and candidate documents are each sampled from human-written summaries and news articles in the CNN/DailyMail dataset (Hermann et al., 2015; See et al., 2017), a common NLP dataset used for summarization task. We select this dataset because the contents are accessible to a general audience, which

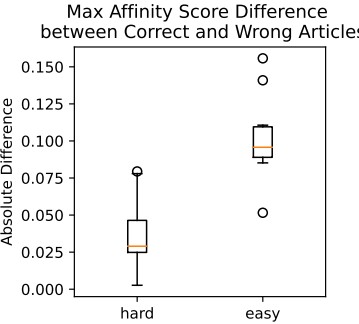

Figure 3: Distribution of affinity scores—computed by the matching model—for hard and easy questions. The box plot shows maximum absolute difference in affinity scores between the correct and wrong candidate articles for each of the hard and easy questions. The smaller the absolute difference is, the smaller the gap between the correct and the wrong article, making the scores less helpful in identifying the correct article (e.g., for the hard questions).

enables us to evaluate a variety of assistive methods by employing crowdworkers as in Lai et al. (2020); Wang & Yin (2021). So in our task, the participants are given a series of questions composed of a query summary with three candidate articles[2], and are asked to select an article from which the summary is generated under a time constraint (Figure 2).

As in the general document matching task, each candidate article is presented with an affinity score computed by a language model, which captures the similarity between the article and the summary. The affinity scores are computed by taking a cosine similarity between the final hidden representation of a language model for the article and the summary (Charlin & Zemel, 2013; Wu et al., 2020). We use the representations from the DistilBART (Shleifer & Rush, 2020) model fine-tuned on the CNN/DailyMail dataset.

**Question types.** In practice, there are some questions where the correct (document) match is obvious, whereas other questions require a more thorough inspection of the specifics. For instance, in scientific peer review, a paper about a new optimization method in deep learning may be assigned to a broad range of candidate reviewers whose general research area is within deep learning. However, a reviewer who has worked both on optimization theory and deep learning may be a better fit compared to others who have primarily worked on large-scale deep-learning based vision models. Even among the reviewers in optimization theory, the reviewers who have worked on similar type of methods to the one proposed may be better suited for the match. Such subtleties require more careful examination by the meta-reviewers.

We capture such scenarios by creating a data pool composed of two types of questions via manual inspection: easy and hard. Easy questions have candidate answers (articles) from different topics or events that are easily discernible from one another, and therefore can be easily matched correctly. On the other hand, hard questions have candidate articles with a shared topic that only differ in small details, requiring a more careful inspection by the users.

On easy versus hard questions, the affinity scores naturally show distinctive behaviors. The gap of the scores between the correct and the wrong matches is smaller for the hard questions than for the easy ones (Figure 3). Because the scores for all candidate articles are similar to one another in the hard questions, the affinity scores are not as helpful in identifying the best match. Additionally, it is more likely that the candidate article with the highest affinity score is not the correct match in the hard questions. If a hypothetical user was to simply select a candidate article which has the highest affinity score by completely relying on the matching model's output, they would be accurate only for 33.3 percent of the time for the hard questions, compared to 100 percent of the time for the easy questions. We believe that providing users with assistive information might be critical for improving outcomes when making decisions on the hard questions, when the model is less accurate and the correct match is more difficult to find.

---

[2]While the decision makers in a general matching task may observe more than three candidate articles, we devise a simpler instantiation here to reduce the complexity of the task, which will be better suited for the crowdsourcing task.

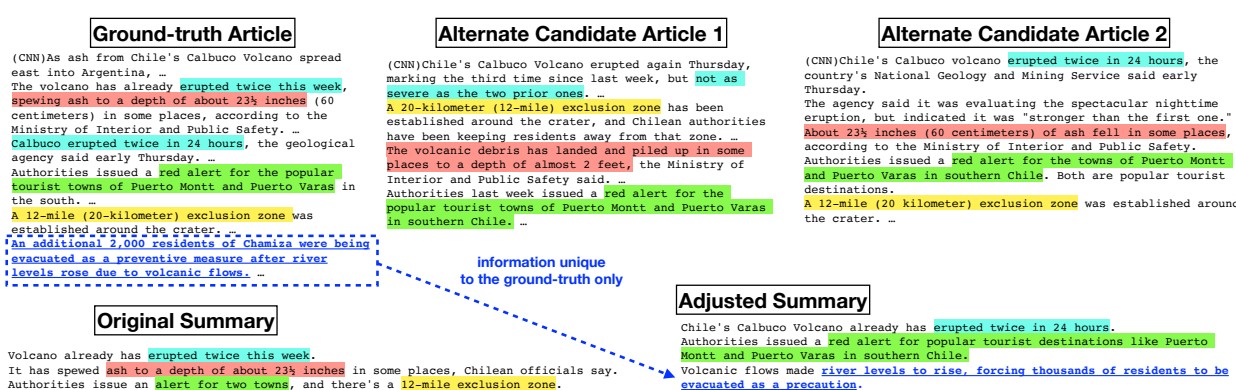

Figure 4: Ensuring a single correct match for the questions. For an original summary sampled from the dataset, the three candidate articles appear to be all correct matches – the critical information highlighted with different colors in the original summary is contained in all three candidate articles (highlighted with the same colors). To resolve having multiple correct matches for the question, we manually extract information unique to the ground-truth article only (underlined text in dotted box) and add it to the adjusted summary to ensure that only the ground-truth article is the correct match for the summary.

**Defining ground-truth matches.** A ground-truth match for a given summary and a set of candidate articles is necessary to measure participant performance. To construct pairs of summary and candidate articles, we first sample a summary-article pair from raw dataset and consider the article as the ground-truth for the given summary. We then select two other articles from the dataset which have the highest affinity scores with respect to the given summary as the incorrect candidate articles for the given summary.

There are several instances where the two alternate candidate articles, which should be incorrect choices, may arguably be a suitable choice for the given summary. This happens because the dataset contains multiple articles covering the same event. To resolve this issue of having multiple ground-truths, we manually modify the given summary so that it is consistent only with the ground-truth article. Specifically, we manually identify unique information in the ground-truth article that is not part of the alternate candidate articles and add that information to the summary (Figure 4).

## 3.2 Tested Methods

In this section, we describe the methods used to highlight assistive information that we evaluate in our study and how they are presented to the users.

**Black-box Model Explanation.** Black-box model explanations include techniques that aim to highlight important input tokens for a model's prediction (Simonyan et al., 2013; Shrikumar et al., 2016; Sundararajan et al., 2017; Lundberg & Lee, 2017). While there are several candidates to consider, we use a widely-applied method called SHAP (Lundberg & Lee, 2017). SHAP assigns attribution scores to each input token that indicate how much they contribute to the prediction output. We select SHAP from a pool of prominent explanation methods (which include Integrated Gradients (Sundararajan et al., 2017) and Input x Gradients (Shrikumar et al., 2016)) by examining how much the distribution of attribution scores deviate from random distribution of attribution scores (see Appendix A for details).

We visualize SHAP (example shown in Figure 5, third row) by highlighting the input tokens according to their attribution scores. Tokens that contribute to increasing the affinity score (i.e., those with positive attribution scores) are highlighted in cyan, while those that decrease the score (i.e., those with negative attribution scores) are highlighted in pink. The color gradients of the highlights indicate the magnitude of the attribution scores: the darker the color, the bigger the magnitude.

**Extractive Summarization.** Summarization methods are trained to select key information within a large body of text. These methods can potentially help users process multiple lengthy articles in a shorter amount of time (Liu & Lapata, 2019; Zhong et al., 2020; Lewis et al., 2020; Zhang et al., 2020). Summaries generated

| Query Summary | Chile's Calbuco Volcano already has erupted twice in 24 hours. Authorities issued a red alert for popular destinations like Puerto Montt and Puerto Varas in southern Chile. Volcanic flows made river levels to rise, forcing thousands of residents to be evacuated as a precaution. |
|---|---|
| Key Parts | (CNN)As ash from Chile's Calbuco Volcano spread east into Argentina, geologists warned of the potential for more activity Friday. Evacuations in the region involved not only people but animals as well. … The volcano has already erupted twice this week, spewing ash to a depth of about 23½ inches (60 centimeters) in some places, according to the Ministry of Interior and Public Safety. … Calbuco erupted twice in 24 hours, the geological agency said early Thursday. … Authorities issued a red alert for the popular tourist towns of Puerto Montt and Puerto Varas in the south. … An additional 2,000 residents of Chamiza were being evacuated as a preventive measure after river levels rose due to volcanic flows. … Another person said: "It was impressive to see an enormous mushroom cloud, with the immense force of the volcano, and to see the ashes. … |
| SHAP | (CNN)As ash from Chile's Calbuco Volcano spread east into Argentina, geologists warned of the potential for more activity Friday. Evacuations in the region involved not only people but animals as well. … The volcano has already erupted twice this week, spewing ash to a depth of about 23½ inches (60 centimeters) in some places, according to the Ministry of Interior and Public Safety. … Calbuco erupted twice in 24 hours, the geological agency said early Thursday. … Authorities issued a red alert for the popular tourist towns of Puerto Montt and Puerto Varas in the south. … An additional 2,000 residents of Chamiza were being evacuated as a preventive measure after river levels rose due to volcanic flows. … Another person said: "It was impressive to see an enormous mushroom cloud, with the immense force of the volcano, and to see the ashes. … |
| BERTSum | (CNN)As ash from Chile's Calbuco Volcano spread east into Argentina, geologists warned of the potential for more activity Friday. Evacuations in the region involved not only people but animals as well. … The volcano has already erupted twice this week, spewing ash to a depth of about 23½ inches (60 centimeters) in some places, according to the Ministry of Interior and Public Safety. … Calbuco erupted twice in 24 hours, the geological agency said early Thursday. … Authorities issued a red alert for the popular tourist towns of Puerto Montt and Puerto Varas in the south. … An additional 2,000 residents of Chamiza were being evacuated as a preventive measure after river levels rose due to volcanic flows. … Another person said: "It was impressive to see an enormous mushroom cloud, with the immense force of the volcano, and to see the ashes. … |
| Co-occurrence | (CNN)As ash from Chile's Calbuco Volcano spread east into Argentina, geologists warned of the potential for more activity Friday. Evacuations in the region involved not only people but animals as well. … The volcano has already erupted twice this week, spewing ash to a depth of about 23½ inches (60 centimeters) in some places, according to the Ministry of Interior and Public Safety. … Calbuco erupted twice in 24 hours, the geological agency said early Thursday. … Authorities issued a red alert for the popular tourist towns of Puerto Montt and Puerto Varas in the south. … An additional 2,000 residents of Chamiza were being evacuated as a preventive measure after river levels rose due to volcanic flows. … Another person said: "It was impressive to see an enormous mushroom cloud, with the immense force of the volcano, and to see the ashes. … |
| Semantic | (CNN)As ash from Chile's Calbuco Volcano spread east into Argentina, geologists warned of the potential for more activity Friday. Evacuations in the region involved not only people but animals as well. … The volcano has already erupted twice this week, spewing ash to a depth of about 23½ inches (60 centimeters) in some places, according to the Ministry of Interior and Public Safety. … Calbuco erupted twice in 24 hours, the geological agency said early Thursday. … Authorities issued a red alert for the popular tourist towns of Puerto Montt and Puerto Varas in the south. … An additional 2,000 residents of Chamiza were being evacuated as a preventive measure after river levels rose due to volcanic flows. … Another person said: "It was impressive to see an enormous mushroom cloud, with the immense force of the volcano, and to see the ashes. … |

Figure 5: Highlighted information using different methods on the ground-truth article of the summary-article pair example in Figure 4. Highlights for "Key Parts" (second row) indicate information relevant to the query summary (first row), all of which ideally should be visibly highlighted by the methods that follow. SHAP (third row) and BERTSum (fourth row) fail to fully highlight all key parts. Critically, they fail to visibly highlight the key part about river levels rising (yellow highlights in Key Parts), the unique information that distinguishes the ground-truth from other candidate articles (as described in Figure 4), which can directly impact the participant's performance. On the other hand, our task-specific methods, both Co-occurrence (fifth row) and Semantic (sixth row) ones, are able to visibly highlight all key parts.

by these methods are typically either abstractive (i.e., the summary is a newly-generated text that may not be part of the original text) or extractive (i.e., the summary is composed of text pieces extracted from the original text) (Hahn & Mani, 2000). Because abstractive summaries are more susceptible to hallucinating information not present in the original text (Cao et al., 2018; Maynez et al., 2020; Ji et al., 2022), we focus on evaluating extractive summaries. In particular, we use BERTSum (Liu & Lapata, 2019), which achieves state-of-the-art performance on the CNN/DailyMail dataset (Pagnoni et al., 2021), to extract three key sentences from the article. We visualize the extracted summary by highlighting the selected sentences from the original text with a single solid color (example shown in Figure 5, fourth row).

**Task-specific Methods.** The summary-article matching task requires users to accurately and quickly identify whether all details in the summary are correctly presented in each article. This is particularly challenging for hard questions, where the ground-truth can only be identified by looking at the right part of the articles due to their subtle differences. Next we propose two *task-specific* methods that are more tailored to addressing this challenge.

The methods operate at sentence and phrase-level information in the summary and candidate articles. Specifically, we select and show the top $K$ sentences[3] from each article with the highest similarity measure to each sentence in the summary[4]. To further provide more fine-grained detail on why that sentence could have been selected, we then show exactly-matching phrases within those selected sentences. Essentially, the methods are designed to guide the users to relevant parts in the article for each summary sentence by presenting the relevance hierarchically – by first showing the key sentences and then the key phrases within.

We consider two versions of the method which use different similarity measures to select the sentences:

- **Co-occurrence method** uses F1 score of ROUGE-L (Lin, 2004), a common performance metric used to capture the degree of n-gram co-occurrence between two texts.

- **Semantic method** uses the cosine similarity between the sentence representations from Sentence-BERT (Reimers & Gurevych, 2019), a transformer model trained for sentence-level tasks. These scores are more sensitive to semantic similarities among texts like paraphrased components that may not be effectively captured by ROUGE-L.

Once we select $K$ sentences based on the similarity measures, we visualize them using different colors to differentiate sentences in the article related to different sentence in the summary. Like before, we use color gradients to indicate the magnitude of the similarity score for each sentence (the higher the similarity, the darker the color). We then color the exactly-matching phrases using the darkest shade. For instance, in Figure 5 (fifth and sixth rows), the pink, blue and yellow highlights indicate relevant parts to first, second, and third sentence in the summary respectively. We include additional examples from each of the explored methods in Appendix C.

## 4 Experiments

We run a pre-registered[5] user study on the summary-article matching task introduced in Section 3.1 to evaluate the methods described in Section 3.2 as treatment conditions. In this section, we outline the details of the user study (Section 4.1), followed by our main hypotheses (Section 4.2) and results (Sections 4.3).

### 4.1 User Study Design

We present 16 questions to each participant. The 16 questions comprise 4 easy and 12 hard questions in random order. Participants complete all questions in one sitting. For each question, participants see a query summary followed by three longer candidate articles (see Figure 2 for an example). To incorporate the time constraints typical decision makers may face in practical settings, as similarly done in (Pier et al., 2017), we limit participants to spend 3 minutes to answer each question, after which they automatically see the next question. We offer bonus payments to encourage high-quality responses in terms of both accuracy and time (more details in Appendix D.4).

We recruit 275 participants from a balanced pool of adult males and females located in the U.S. with minimum approval ratings of 90% on Prolific (`www.prolific.co`), with diverse demographic background (more details in Appendix D.2). The sample size is determined from Monte Carlo power analysis based on data collected from a separate pilot study, for a statistical power above 0.8 (more details in Appendix D.1). Each participant is then randomly assigned to one of five groups:

---

[3]We pick $K = 3$, but this can be tuned for different levels of detail, depending on the length of the summary or the article.

[4]Note that one could alternatively consider applying SHAP to the sentence-level information from the summary instead of the entire summary. While seemingly providing an apples-to-apples comparison to the sentence-level task-specific methods, a sentence-level application of SHAP would be unconventional and inconsistent with SHAP's intended use. Indeed, SHAP is designed to explain a model's prediction, and in our set up the model makes a prediction for the affinity score between the *entire summary* and the article. Hence, the most natural and consistent way of using SHAP is applying it to the entire input. For text summarization method, a similar argument holds: the method summarizes a longer text by extracting a subset of information and therefore is meant to take in the entire document content as the input.

[5]Pre-registration document is available at `https://aspredicted.org/LMM_4K9`

- *Control*: participants see the basic information (summary, articles, affinity scores)

- *SHAP*: participants see the basic information + highlights from SHAP

- *BERTSum*: participants see the basic information + highlights from BERTSum

- *Co-occurrence*: participants see the basic information + highlights from Co-occurrence method

- *Semantic*: participants see the basic information + highlights from Semantic method

We include two attention check questions in the study in addition to the 16 questions above. 271 out of 275 participants pass both attention-check questions, and we exclude responses from the 4 non-qualifying participants from our further analysis. We include mode details about the user study in Appendix D.

## 4.2 Main Hypotheses

We pose the following null hypotheses with two-sided alternatives about the mean accuracy of participants on the hard questions,[6] using different kinds of assistive information:

(**H1**) The mean accuracy of participants using SHAP is not different from that of the control.

(**H2**) The mean accuracy of participants using BERTSum is not different from that of the control.

(**H3**) The mean accuracy of participants using Co-occurrence method is not different from the control.

(**H4**) The mean accuracy of participants using Semantic method is not different from the control.

To test each hypothesis, we compare the mean accuracy of the participants in different treatment settings against the control group with two-tailed permutation tests, where the test statistic is the difference in the mean accuracy. We account for multiple comparisons with Sidak correction (Sidak, 1967) for the family-wise error rate of 0.05.

## 4.3 Results

We now discuss the participants' task accuracy (Section 4.3.1), completion time (Section 4.3.2), and qualitative responses (Section 4.3.3) in different treatment groups.

### 4.3.1 Accuracy Difference

We find a statistically significant difference in the mean accuracy of all treatment groups compared to the control and reject the null hypotheses **H1** through **H4** in Section 4.2.

- Participants using SHAP perform significantly *worse* than the control ($p = 0.001599 < 0.05$).

- Participants using BERTSum perform significantly *worse* than the control ($p = 0.0056 < 0.05$).

- Participants using Co-occurrence method perform significantly *better* than the control ($p = 0.002997 < 0.05$).

- Participants using using the Semantic method perform significantly *better* than the control ($p = 0.002997 < 0.05$).

---

[6]We do not include statistical tests for the easy questions because they were not part of the pre-registered experiment. This was for two reasons. First, if the tests for the easy questions were to be included, the effective sample size would have become much larger to draw reliable conclusions about the methods (taking multiple testing into account), increasing the cost of the experiment significantly. Second, we were more interested in the methods' ability to assist with the hard questions as such situation is more likely to call for additional assistive information. Hence, we prioritized testing the hard questions.

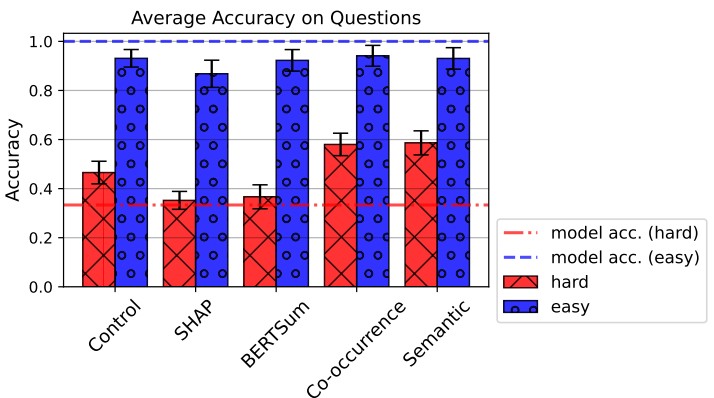

(a) Average accuracy with 95% confidence intervals.

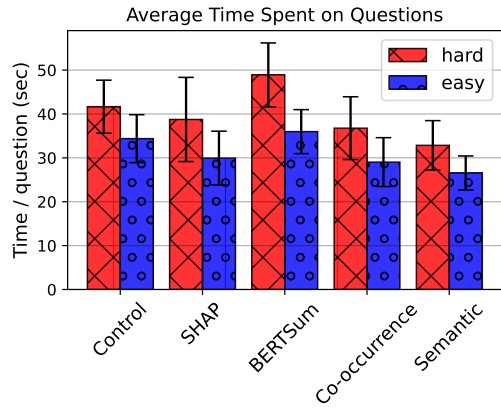

(b) Average time with 95% confidence intervals.

Figure 6: (a) On hard questions, we observe significantly higher accuracy in groups presented with assistive information from Co-occurrence and Semantic methods compared to the control, and lower accuracy in groups presented with assistive information from SHAP and BERTSum. The dotted lines indicate the accuracy of the matching model, i.e., accuracy when selecting the article with the highest affinity score. For easy questions, it is more effective to simply follow the affinity scores without support from additional assistive information. For hard questions where the correct match is less obvious, using Co-occurrence or Semantic methods may be effective. (b) We observe a lower average time for groups given SHAP, Co-occurrence, and Semantic methods compared to control in both types of questions, but higher average time for BERTSum.

Comparing the participants' accuracy against the model accuracy on different question types, we verify that the assistive information is particularly useful for the hard questions (Figure 6a). Note that the model is only accurate around 33.3% of the time in hard questions (red dotted line). The information from Semantic method is the most effective for the hard questions with the highest average accuracy of 58.6%, which is a 26% increase in accuracy compared to the control (46.6%) and a 77% increase compared to the model accuracy (33.3%), while SHAP (35.2%) and BERTSum (36.7%) remain less effective (red checker-patterned bars).

On the other hand, there appears no significant difference in accuracy among the methods on the easy questions (blue dotted bars), all of them slightly less accurate than the model accuracy (blue dotted line). Highly-overlapping confidence intervals among all methods and the control also imply that none of the methods tested may be particularly useful on the easy questions compared to the control. The diverging behaviors of the same methods in easy and hard questions suggest different regimes in which the methods can be particularly effective: while it is more efficient to simply rely on the affinity scores for the easy questions, assistive information (via Semantic method) can be particularly helpful for the hard questions, when the correct match is less obvious for both the models and humans. This further suggests that for the best results in practice, it may be useful to consider first identifying the difficulty of the question and then deciding if additional information should be provided.

### 4.3.2   Time Difference

We record the average response time (in seconds/question) for participants in each treatment group. We observe that on average the participants using SHAP, Co-occurrence, and Semantic methods respond more quickly compared to the control group for both types of questions (Figure 6b). For both easy and hard questions, the participants using the Semantic method take the shortest average time (26.6 seconds for easy and 32.9 seconds for hard), which is approximately a 20% improvement over the control (34.4 seconds for easy and 41.7 seconds for hard). The participants using BERTSum take the longest (36 seconds for easy and 48.9 seconds for hard), where they experience a 17% increase in time for the hard questions. Given that the Semantic method is also able to significantly boost the task accuracy, it is the most effective method among

the tested ones. On the other hand, as BERTSum simultaneously decreases the task accuracy and increases the completion time, it may be considered the least effective method.

### 4.3.3 Qualitative Responses

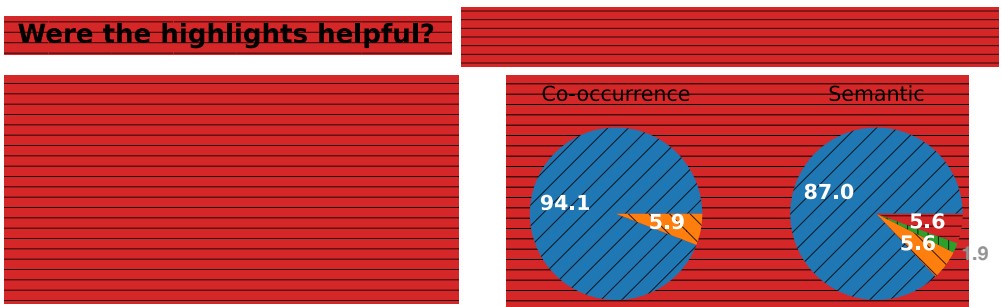

Figure 7: Participants' responses (in percentage) for "Were the highlights helpful?". For all the methods, the majority of the participants respond positively to the question regardless of their actual task accuracy.

At the end of the user study, the participants are asked several qualitative questions about the task.

*"Were the highlights helpful?"* Participants from all of the treatment groups generally respond positively to this question – Figure 7 shows the proportion of different responses from the participants in each group, and positive responses in blue form the majority in all groups. While the participants *believe* the highlights to be helpful, their task performance shows the contrary for participants using SHAP and BERTSum. Such discrepancy between the subjective perception of a tool's utility and the objective utility measured by task-grounded performance metrics corroborate similar previous observations on different assistive tools (Kaur et al., 2020; Bansal et al., 2021).

*"What information was the most helpful in answering the question?"* While the majority of the participants using BERTSum, Co-occurrence, and Semantic methods respond that the highlights were the most helpful, the participants using SHAP have more diverse responses that showed no particular preference (Figure 8). It is interesting to note that the participants using either Co-occurrence or Semantic methods find the role of highlights to be significantly helpful when compared to other methods.

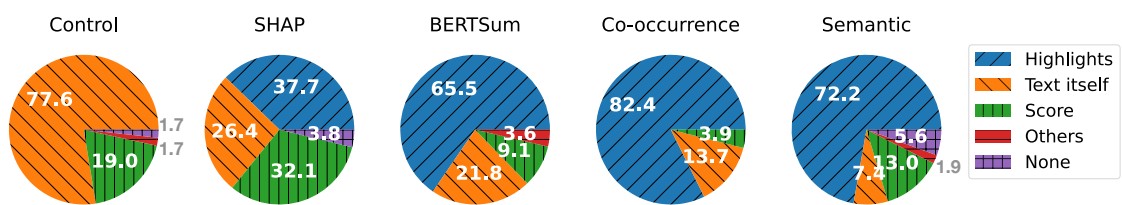

Figure 8: Participants' responses (in percentage) to a question "What information was the most helpful?"

*"Were there too many highlights?"* We find that the participants using SHAP most strongly agree to this sentiment (Figure 9). One factor that could have contributed to this is the default output values from SHAP used to generate the highlights, which are not post-processed for more succinct representation of information. Appropriate post-processing of the attribution scores may be necessary to better account for this—the impact of the amount of highlights on the task performance is an open research question that requires future work.

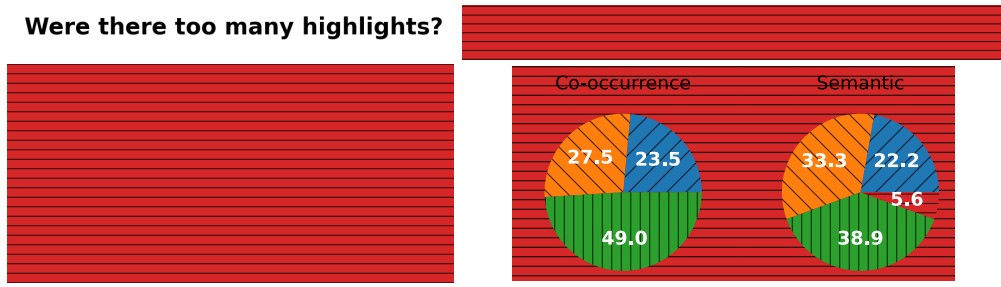

Figure 9: Participants' responses (in percentage) to a question "Were there too many highlights?"

## 5 Conclusion

Motivated by practical concerns in document matching tasks with human decision makers, we conducted a user study to investigate the utility of different kinds of assistive information for the summary-article matching task. We found that even well-established black-box model explanations can potentially impair the users' decisions, while task-specific approaches can effectively assist them. Existing methods are typically not explicitly optimized for the task's objective: Model explanations are contingent on the matching model; it attempts to explain what the *model* considers important, not necessarily what *human* users find important to perform well in the task. This misalignment can be particularly confusing to the users if the model prediction is wrong (e.g., as observed in hard questions of the summary-article matching) where the explanations are more of "justification" of why the prediction is wrong. General text summarization methods can be helpful for succinctly expressing a high-level topic of the article, but may lack the precision of picking the details directly related to the given summary. Furthermore, we observed that the users' subjective perception on the utility of (assistive) information was misaligned with their performance on the task. These results altogether emphasize that it is important for the developers of such assistive tools to articulate the specific use (and users) it serves, and rigorously evaluate their proposals.

We believe that the summary-article matching task can be used as a first-pass test to validate promising methods (and promote development of new approaches). Relaxing some assumptions in our setup can provide further insights on strengths and weaknesses of individual methods in more complex scenarios (e.g., allowing multiple or no ground-truths in questions). Complex and lengthy documents in practice (e.g., academic papers and reviewer profiles) may require longer data collection phase and more scalable adaptations of the presented methods. For instance, visualizing all highlights with different colors at once (as we did in the summary-article matching task) may not be practical for longer documents. Necessary changes in the user interface can address such issues, e.g., the highlights can be shown interactively based on what information the user is interested in. We present preliminary details of how we adapt this tool[7] for a peer review use case in Appendix E with some initial feedback. It could also be valuable to explore ways to improve the affinity scoring model itself, potentially utilizing the extracted information presented by these methods.

### Broader Impact Statement

The assistive methods should be subject to further experiments before being applied to domains with major social implications, e.g., hiring. The methods should also be designed and revised to abide by any regulatory guidelines within. The methods' output also should not serve as the sole justification for confirming a match in practice, as it is only intended as a *supplementary* information that the users may use at their discretion. The correctness of the output of the method is not guaranteed and can vary depending on the internal models used, which require further independent research and experimentation.

---

[7]https://huggingface.co/spaces/jskim/paper-matching

**Acknowledgments**

We thank Kasun Amarasinghe, Siddhant Arora, Keegan Harris, Nari Johnson, and Yilun Zhou for helpful feedback and discussions. This research was approved by CMU Institutional Review Board (IRB). This work was supported in part by the National Science Foundation grants CIF1942124, IIS1705121, IIS1838017, IIS2046613, IIS2112471, CIF1763734 and funding from Meta, Morgan Stanley, Amazon, and Google. Any opinions, findings and conclusions or recommendations expressed in this material are those of the author(s) and do not necessarily reflect the views of any of these funding agencies.

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

# Appendices

## A   Proxy Test of Black-box Model Explanations

There are several black-box model explanations to consider for the task. While testing all of them on real users can be an interesting research on its own, as we are more broadly interested in what distinct types of information could be helpful, we decide to select one representative method in the literature. As there is no absolute answer to which method is superior, we conduct a simple proxy test of what method can be a better choice for the task.

We consider the following feature attribution methods: Integrated Gradients (Sundararajan et al., 2017), Input x Gradients (Shrikumar et al., 2016), and SHAP (Lundberg & Lee, 2017). In Figure 10, we plot the mean EM distance (averaged across 50 different random attributions, normalized to be between 0 and 1) between the distribution of attribution scores for the input tokens in our ground-truth articles. The higher the value (darker the color), the more distinct the distribution of the attribution scores computed by respective methods. Notice that SHAP shows the most distinct distribution from random attributions compared to other methods, indicating it may be a better choice that carry more information about the important tokens. We have also qualitatively verified that the highlights from other two methods were not as meaningful as SHAP on the articles.

Note also that SHAP is a promising candidate to apply to the task due to its popularity and its common presence in more sophisticated domains like biology, physics, chemistry, and finance (Jesus et al., 2021; Novakovsky et al., 2022; Yang et al., 2022; Zablocki et al., 2022; Pucci et al., 2022).

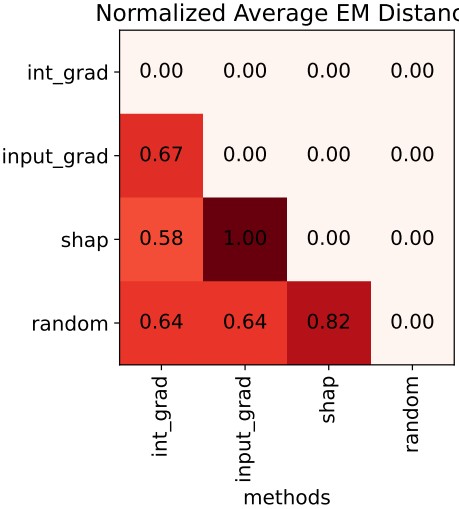

Figure 10: Proxy quality test for the black-box model explanations using average EM distances between the distributions of attribution scores of input tokens. The higher the values (the darker the color), the more different the distribution of the attribution scores. SHAP shows the most distinct distribution from the random attributions (bottom row).

# B    Method Details

## B.1    SHAP

We use the native implementation of the method (`https://github.com/slundberg/shap`), version 0.40.0, and use the default hyperparameters set by the library. We apply SHAP to the matching model, which takes in the summary text and one candidate article to compute the affinity score. The matching model uses the embeddings from DistilBART(`https://huggingface.co/sshleifer/distilbart-cnn-12-6`) fine-tuned on the same CNN/DailyMail dataset, and computes the cosine similarity between the embeddings.

## B.2    BertSUM

We use the native implementation of the method (`https://github.com/nlpyang/PreSumm`) and apply it directly to the candidate articles. In particular, we use the extractive model in the repository that has been fine-tuned on the same CNN/DailyMail dataset (`BertSUMExt`). We use the default hyperparameters set by the original source code, extracting three sentences for the summary.

## B.3    Co-occurrence Method

We use `NLTK`(`https://www.nltk.org/`) library to first tokenize the candidate articles into different sentences. Then we use the Python package `rouge-score`(`https://pypi.org/project/rouge-score/`) to compute the sentence-wise ROUGE-L F1 scores. These scores are used to highlight the sentences. We assign different colors of highlights to parts that are relevant to the different sentences in the summary.

## B.4    Semantic Method

Similar to the co-occurrence method, we first tokenize the candidate articles into sentences. Then we use `sentence-transformers`(`https://www.sbert.net/index.html`) library version 2.2.2 (model used: `all-MiniLM-L6-v2`) to obtain sentence embeddings, which are subsequently used to compute the sentence-level affinity scores. These scores are then used to highlight the sentences. The highlighting scheme remains the same as the co-occurrence method.

# C   Method Examples

We show below some example highlights presented to the users using different methods.

**Summary**

The United Nations Relief and Works Agency chief will visit Yarmouk camp Saturday . Militant groups are currently in control of the camp . Yarmouk has been engulfed in fighting since December 2012 .

**Article 1 --- Score: 0.75**

CNN) The commissioner - general of the United Nations Relief and Works Agency will make an emergency visit to the Yarmouk Palestinian refugee camp in Syria on Saturday, a spokesman says. Commissioner - General Pierre Krähenbühl will assess the humanitarian situation in the camp and speak with individuals about ways to relieve the suffering of the people who remain there. "The visit is prompted by UNRWA's deepening concern for the safety and protection of 18, 000 Palestinians and Syrian civilians, including 3, 500 children," agency spokesman Christopher Gunness told CNN's Paula Newton. "Yarmouk remains under the control of armed groups, and civilian life continues to be threatened by the effects of the conflict." Krähenbühl will meet with senior Syrian officials, U. N. and relief agency staff members, and displaced people from the camp itself. The Yarmouk refugee camp, which sits just 6 miles from central Damascus, has been engulfed in fighting between the Syrian government and armed groups since December 2012. The London - based Syrian Observatory for Human Rights says the militant group ISIS and the al Qaeda - affiliated Al - Nusra Front control about 90% of the camp. The organization also claims that the Syrian government has dropped barrel bombs on the camp as recently as Sunday in an effort to drive out armed groups. Yarmouk was formed in 1957 to accommodate people displaced by the Arab - Israeli conflict and is the largest Palestinian refugee camp in Syria. The U. N. relief agency estimates that there were 160, 000 people in the camp when the conflict began in 2011 between forces loyal to President Bashar al - Assad and opposition fighters. That number has dropped to about 18, 000, according to estimates. Yarmouk has been largely cut off from aid since November 2013. There have been widespread reports of malnutrition and shortages of medical care. "We will not abandon hope," Gunness said. "We will not submit to pessimism, because to abandon hope would be to abandon the people of Yarmouk. ... We cannot abandon the people of Yarmouk, and we will not, hence this mission."

**Article 2 --- Score: 0.73**

CNN) Thousands of Palestinians are trapped in the devastated Yarmouk refugee camp in Syria, which has mostly been seized by groups including ISIS, activists report. The London - based Syrian Observatory for Human Rights says ISIS and the al Qaeda - affiliated Al - Nusra Front took control of 90% of the camp in southern Damascus. Calling the lives of Yarmouk refugees "profoundly threatened "on Sunday, the United Nations Relief and Works Agency issued a statement urging humanitarian aid access. "Never has the hour been more desperate in the Palestine refugee camp of Yarmouk," the statement said. The UNRWA estimates 18, 000 civilians remain trapped in the camp that has been engulfed in fighting between the government and rebel forces since December 2012. Syria's state - run SANA news agency reports up to 2, 000 people have fled in the past two days as food, water and medical supplies remain scarce. "All people are trying to leave the camp," says Syrian activist Abu Mohammed in Damascus who used to live in Yarmouk. "There is no electricity," says Mohammed. "ISIS controls the hospital so injured people have nowhere to go." The Syrian Observatory for Human Rights reports barrel bombs were dropped on the camp Sunday as clashes continued. The Palestine Liberation Organization called on international bodies to assist in the evacuation of people from the camp. "Reports of kidnappings, beheadings and mass killings are coming out from Al - Yarmouk, which is under a brutal campaign of murder and occupation," Palestine Liberation Organization Executive Committee Member Dr. Saeb Erekat said Saturday. Yarmouk, the largest Palestinian refugee camp in Syria, was formed in 1957 to accommodate people fleeing the Arab - Israeli conflict. "The levels of humanity that we have seen have now descended into further levels of inhumanity," said Chris Gunness, spokesman for the UNRWA. Yarmouk, he added, "was always a place where human rights meant very little. We are seeing it descend further." CNN's Samira Said contributed to this report.

**Article 3 --- Score: 0.72**

CNN) They took Yarmouk by storm, a sea of masked men flooding into the streets of one the world's most beleaguered places. Besieged and bombed by Syrian forces for more than two years, the desperate residents of this Palestinian refugee camp near Damascus awoke in early April to a new, even more terrifying reality -- ISIS militants seizing Yarmouk after defeating several militia groups operating in the area. "They slaughtered them in the streets," one Yarmouk resident, who asked not to be named, told CNN. "They caught) three people and killed them in the street, in front of people. The Islamic State is now in control of almost all the camp." An estimated 18, 000 refugees are now trapped inside Yarmouk, stuck between ISIS and Syrian regime forces in "the deepest circle of hell," in the words of U. N. Secretary - General Ban Ki - moon. Yarmouk, the largest Palestinian refugee camp in Syria, was formed in 1957 to accommodate people fleeing the Arab - Israeli conflict. The camp, which sits just 6 miles from central Damascus, has been engulfed in fighting between the Syrian government and armed groups since December 2012. The London - based Syrian Observatory for Human Rights says ISIS and the al Qaeda - affiliated Al - Nusra Front control about 90% of the camp. The organization also claims that the Syrian government has dropped barrel bombs on the camp in an effort to drive out armed groups. Activists and residents in Yarmouk tell CNN that as many as 5, 000 people have tried to flee their homes since ISIS stormed the camp, but have no place to go. Hundreds have been injured, but the camp's only functioning hospital was first occupied by ISIS, then targeted last week by regime shelling. As the fighting raged in Yarmouk, the director of the Jafra Foundation -- the only aid group that has been able to get into the camp -- painted a grim portrait of the conditions on the ground since ISIS arrived. "We need medicine and access to treatment and medical facilities," Wesam Sabaneh told CNN. "The last hospital in Yarmouk camp was bombed yesterday, so there's really nothing functioning." Opinion: Save the ' miracle babies ' Even delivering clean water in Yarmouk can be a deadly task. Majed Alomari, the Jafra Foundation's water coordinator, was killed a few days ago -- gunned down in an ISIS firefight with rival rebel groups. The head of the Palestinian League for Human Rights in Syria

Figure 11: Example highlights for SHAP.

**Summary**

The United Nations Relief and Works Agency chief will visit Yarmouk camp Saturday . Militant groups are currently in control of the camp . Yarmouk has been engulfed in fighting since December 2012 .

**Article 1 --- Score: 0.75**

(CNN)The commissioner-general of the United Nations Relief and Works Agency will make an emergency visit to the Yarmouk Palestinian refugee camp in Syria on Saturday, a spokesman says. Commissioner-General Pierre Krähenbühl will assess the humanitarian situation in the camp and speak with individuals about ways to relieve the suffering of the people who remain there. "The visit is prompted by UNRWA's deepening concern for the safety and protection of 18,000 Palestinians and Syrian civilians, including 3,500 children," agency spokesman Christopher Gunness told CNN's Paula Newton. "Yarmouk remains under the control of armed groups, and civilian life continues to be threatened by the effects of the conflict." Krähenbühl will meet with senior Syrian officials, U.N. and relief agency staff members, and displaced people from the camp itself. The Yarmouk refugee camp, which sits just 6 miles from central Damascus, has been engulfed in fighting between the Syrian government and armed groups since December 2012. The London-based Syrian Observatory for Human Rights says the militant group ISIS and the al Qaeda-affiliated Al-Nusra Front control about 90% of the camp. The organization also claims that the Syrian government has dropped barrel bombs on the camp as recently as Sunday in an effort to drive out armed groups. Yarmouk was formed in 1957 to accommodate people displaced by the Arab-Israeli conflict and is the largest Palestinian refugee camp in Syria. The U.N. relief agency estimates that there were 160,000 people in the camp when the conflict began in 2011 between forces loyal to President Bashar al-Assad and opposition fighters. That number has dropped to about 18,000, according to estimates. Yarmouk has been largely cut off from aid since November 2013. There have been widespread reports of malnutrition and shortages of medical care. "We will not abandon hope," Gunness said. "We will not submit to pessimism, because to abandon hope would be to abandon the people of Yarmouk. ... We cannot abandon the people of Yarmouk, and we will not, hence this mission."

**Article 2 --- Score: 0.73**

(CNN)Thousands of Palestinians are trapped in the devastated Yarmouk refugee camp in Syria, which has mostly been seized by groups including ISIS, activists report. The London-based Syrian Observatory for Human Rights says ISIS and the al Qaeda-affiliated Al-Nusra Front took control of 90% of the camp in southern Damascus. Calling the lives of Yarmouk refugees "profoundly threatened" on Sunday, the United Nations Relief and Works Agency issued a statement urging humanitarian aid access. "Never has the hour been more desperate in the Palestine refugee camp of Yarmouk," the statement said. The UNRWA estimates 18,000 civilians remain trapped in the camp that has been engulfed in fighting between the government and rebel forces since December 2012. Syria's state-run SANA news agency reports up to 2,000 people have fled in the past two days as food, water and medical supplies remain scarce. "All people are trying to leave the camp," says Syrian activist Abu Mohammed in Damascus who used to live in Yarmouk. "There is no electricity," says Mohammed. "ISIS controls the hospital so injured people have nowhere to go." The Syrian Observatory for Human Rights reports barrel bombs were dropped on the camp Sunday as clashes continued. The Palestine Liberation Organization called on international bodies to assist in the evacuation of people from the camp. "Reports of kidnappings, beheadings and mass killings are coming out from Al- Yarmouk, which is under a brutal campaign of murder and occupation," Palestine Liberation Organization Executive Committee Member Dr. Saeb Erekat said Saturday. Yarmouk, the largest Palestinian refugee camp in Syria, was formed in 1957 to accommodate people fleeing the Arab-Israeli conflict. "The levels of humanity that we have seen have now descended into further levels of inhumanity," said Chris Gunness, spokesman for the UNRWA. Yarmouk, he added, "was always a place where human rights meant very little. We are seeing it descend further." CNN's Samira Said contributed to this report .

**Article 3 --- Score: 0.72**

(CNN)They took Yarmouk by storm, a sea of masked men flooding into the streets of one the world's most beleaguered places. Besieged and bombed by Syrian forces for more than two years, the desperate residents of this Palestinian refugee camp near Damascus awoke in early April to a new, even more terrifying reality -- ISIS militants seizing Yarmouk after defeating several militia groups operating in the area. "They slaughtered them in the streets," one Yarmouk resident, who asked not to be named, told CNN. "They (caught) three people and killed them in the street, in front of people. The Islamic State is now in control of almost all the camp." An estimated 18,000 refugees are now trapped inside Yarmouk, stuck between ISIS and Syrian regime forces in "the deepest circle of hell," in the words of U.N. Secretary-General Ban Ki-moon. Yarmouk, the largest Palestinian refugee camp in Syria, was formed in 1957 to accommodate people fleeing the Arab-Israeli conflict. The camp, which sits just 6 miles from central Damascus, has been engulfed in fighting between the Syrian government and armed groups since December 2012. The London-based Syrian Observatory for Human Rights says ISIS and the al Qaeda-affiliated Al-Nusra Front control about 90% of the camp. The organization also claims that the Syrian government has dropped barrel bombs on the camp in an effort to drive out armed groups. Activists and residents in Yarmouk tell CNN that as many as 5,000 people have tried to flee their homes since ISIS stormed the camp, but have no place to go. Hundreds have been injured, but the camp's only functioning hospital was first occupied by ISIS, then targeted last week by regime shelling. As the fighting raged in Yarmouk, the director of the Jafra Foundation -- the only aid group that has been able to get into the camp -- painted a grim portrait of the conditions on the ground since ISIS arrived. "We need medicine and access to treatment and medical facilities," Wesam Sabaneh told CNN. "The last hospital in Yarmouk camp was bombed yesterday, so there's really nothing functioning." Opinion: Save the 'miracle babies' Even delivering clean water in Yarmouk can be a deadly task. Majed Alomari, the Jafra Foundation's water coordinator, was killed a few days ago -- gunned down in an ISIS firefight with rival rebel groups. The head of the Palestinian League for Human Rights in Syria (PLHR), who fled the camp and Syria in October 2012, said the people of Yarmouk were in dire need of help.

Figure 12: Example highlights for BERTSum.

**Summary**

The United Nations Relief and Works Agency chief will visit Yarmouk camp Saturday . Militant groups are currently in control of the camp . Yarmouk has been engulfed in fighting since December 2012 .

**Article 1 --- Score: 0.75**

( CNN ) The commissioner - general of the United Nations Relief and Works Agency will make an emergency visit to the Yarmouk Palestinian refugee camp in Syria on Saturday , a spokesman says . Commissioner - General Pierre Krähenbühl will assess the humanitarian situation in the camp and speak with individuals about ways to relieve the suffering of the people who remain there . `` The visit is prompted by UNRWA 's deepening concern for the safety and protection of 18,000 Palestinians and Syrian civilians , including 3,500 children , " agency spokesman Christopher Gunness told CNN 's Paula Newton . `` Yarmouk remains under the control of armed groups , and civilian life continues to be threatened by the effects of the conflict . " Krähenbühl will meet with senior Syrian officials , U.N. and relief agency staff members , and displaced people from the camp itself . The Yarmouk refugee camp , which sits just 6 miles from central Damascus , has been engulfed in fighting between the Syrian government and armed groups since December 2012 . The London - based Syrian Observatory for Human Rights says the militant group ISIS and the al Qaeda - affiliated Al - Nusra Front control about 90 % of the camp . The organization also claims that the Syrian government has dropped barrel bombs on the camp as recently as Sunday in an effort to drive out armed groups . Yarmouk was formed in 1957 to accommodate people displaced by the Arab - Israeli conflict and is the largest Palestinian refugee camp in Syria . The U.N. relief agency estimates that there were 160,000 people in the camp when the conflict began in 2011 between forces loyal to President Bashar al - Assad and opposition fighters . That number has dropped to about 18,000 , according to estimates . Yarmouk has been largely cut off from aid since November 2013 . There have been widespread reports of malnutrition and shortages of medical care . `` We will not abandon hope , " Gunness said . `` We will not submit to pessimism , because to abandon hope would be to abandon the people of Yarmouk . ... We can not abandon the people of Yarmouk , and we will not , hence this mission . "

**Article 2 --- Score: 0.73**

( CNN ) Thousands of Palestinians are trapped in the devastated Yarmouk refugee camp in Syria , which has mostly been seized by groups including ISIS , activists report . The London - based Syrian Observatory for Human Rights says ISIS and the al Qaeda - affiliated Al - Nusra Front took control of 90 % of the camp in southern Damascus . Calling the lives of Yarmouk refugees `` profoundly threatened " on Sunday , the United Nations Relief and Works Agency issued a statement urging humanitarian aid access . `` Never has the hour been more desperate in the Palestine refugee camp of Yarmouk , " the statement said . The UNRWA estimates 18,000 civilians remain trapped in the camp that has been engulfed in fighting between the government and rebel forces since December 2012 . Syria 's state - run SANA news agency reports up to 2,000 people have fled in the past two days as food , water and medical supplies remain scarce . `` All people are trying to leave the camp , " says Syrian activist Abu Mohammed in Damascus who used to live in Yarmouk . `` There is no electricity , " says Mohammed . `` ISIS controls the hospital so injured people have nowhere to go . " The Syrian Observatory for Human Rights reports barrel bombs were dropped on the camp Sunday as clashes continued . The Palestine Liberation Organization called on international bodies to assist in the evacuation of people from the camp . `` Reports of kidnappings , beheadings and mass killings are coming out from Al - Yarmouk , which is under a brutal campaign of murder and occupation , " Palestine Liberation Organization Executive Committee Member Dr. Saeb Erekat said Saturday . Yarmouk , the largest Palestinian refugee camp in Syria , was formed in 1957 to accommodate people fleeing the Arab - Israeli conflict . `` The levels of humanity that we have seen have now descended into further levels of inhumanity , " said Chris Gunness , spokesman for the UNRWA . Yarmouk , he added , `` was always a place where human rights meant very little . We are seeing it descend further . " CNN 's Samira Said contributed to this report .

**Article 3 --- Score: 0.72**

( CNN ) They took Yarmouk by storm , a sea of masked men flooding into the streets of one the world 's most beleaguered places . Besieged and bombed by Syrian forces for more than two years , the desperate residents of this Palestinian refugee camp near Damascus awoke in early April to a new , even more terrifying reality -- ISIS militants seizing Yarmouk after defeating several militia groups operating in the area . `` They slaughtered them in the streets , " one Yarmouk resident , who asked not to be named , told CNN . `` They ( caught ) three people and killed them in the street , in front of people . The Islamic State is now in control of almost all the camp . " An estimated 18,000 refugees are now trapped inside Yarmouk , stuck between ISIS and Syrian regime forces in `` the deepest circle of hell , " in the words of U.N. Secretary - General Ban Ki - moon . Yarmouk , the largest Palestinian refugee camp in Syria , was formed in 1957 to accommodate people fleeing the Arab - Israeli conflict . The camp , which sits just 6 miles from central Damascus , has been engulfed in fighting between the Syrian government and armed groups since December 2012 . The London - based Syrian Observatory for Human Rights says ISIS and the al Qaeda - affiliated Al - Nusra Front control about 90 % of the camp . The organization also claims that the Syrian government has dropped barrel bombs on the camp in an effort to drive out armed groups . Activists and residents in Yarmouk tell CNN that as many as 5,000 people have tried to flee their homes since ISIS stormed the camp , but have no place to go . Hundreds have been injured , but the camp 's only functioning hospital was first occupied by ISIS , then targeted last week by regime shelling . As the fighting raged in Yarmouk , the director of the Jafra Foundation -- the only aid group that has been able to get into the camp -- painted a grim portrait of the conditions on the ground since ISIS arrived . `` We need medicine and access to treatment and medical facilities , " Wesam Sabaneh told CNN . `` The last hospital in Yarmouk camp was bombed yesterday , so there 's really nothing functioning . " Opinion : Save the 'miracle babies ' Even delivering clean water in Yarmouk can be a deadly task . Majed Alomari , the Jafra Foundation 's water coordinator , was killed a few days ago -- gunned down in an ISIS firefight with rival rebel groups . The head of the Palestinian League for Human Rights in Syria ( PLHR ) , who fled the camp and Syria in October 2012 , said the people of Yarmouk were in dire need of help .

Figure 13: Example highlights for Co-occurrence method.

**Summary**

The United Nations Relief and Works Agency chief will visit Yarmouk camp Saturday . Militant groups are currently in control of the camp . Yarmouk has been engulfed in fighting since December 2012 .

**Article 1 --- Score: 0.75**

( CNN ) The commissioner - general of the United Nations Relief and Works Agency will make an emergency visit to the Yarmouk Palestinian refugee camp in Syria on Saturday , a spokesman says . Commissioner - General Pierre Krähenbühl will assess the humanitarian situation in the camp and speak with individuals about ways to relieve the suffering of the people who remain there . `` The visit is prompted by UNRWA 's deepening concern for the safety and protection of 18,000 Palestinians and Syrian civilians , including 3,500 children , " agency spokesman Christopher Gunness told CNN 's Paula Newton . `` Yarmouk remains under the control of armed groups , and civilian life continues to be threatened by the effects of the conflict . " Krähenbühl will meet with senior Syrian officials , U.N. and relief agency staff members , and displaced people from the camp itself . The Yarmouk refugee camp , which sits just 6 miles from central Damascus , has been engulfed in fighting between the Syrian government and armed groups since December 2012 . The London - based Syrian Observatory for Human Rights says the militant group ISIS and the al Qaeda - affiliated Al - Nusra Front control about 90 % of the camp . The organization also claims that the Syrian government has dropped barrel bombs on the camp as recently as Sunday in an effort to drive out armed groups . Yarmouk was formed in 1957 to accommodate people displaced by the Arab - Israeli conflict and is the largest Palestinian refugee camp in Syria . The U.N. relief agency estimates that there were 160,000 people in the camp when the conflict began in 2011 between forces loyal to President Bashar al - Assad and opposition fighters . That number has dropped to about 18,000 , according to estimates . Yarmouk has been largely cut off from aid since November 2013 . There have been widespread reports of malnutrition and shortages of medical care . `` We will not abandon hope , " Gunness said . `` We will not submit to pessimism , because to abandon hope would be to abandon the people of Yarmouk . ... We can not abandon the people of Yarmouk , and we will not , hence this mission . "

**Article 2 --- Score: 0.73**

( CNN ) Thousands of Palestinians are trapped in the devastated Yarmouk refugee camp in Syria , which has mostly been seized by groups including ISIS , activists report . The London - based Syrian Observatory for Human Rights says ISIS and the al Qaeda - affiliated Al - Nusra Front took control of 90 % of the camp in southern Damascus . Calling the lives of Yarmouk refugees `` profoundly threatened " on Sunday , the United Nations Relief and Works Agency issued a statement urging humanitarian aid access . `` Never has the hour been more desperate in the Palestine refugee camp of Yarmouk , " the statement said . The UNRWA estimates 18,000 civilians remain trapped in the camp that has been engulfed in fighting between the government and rebel forces since December 2012 . Syria 's state - run SANA news agency reports up to 2,000 people have fled in the past two days as food , water and medical supplies remain scarce . `` All people are trying to leave the camp , " says Syrian activist Abu Mohammed in Damascus who used to live in Yarmouk . `` There is no electricity , " says Mohammed . `` ISIS controls the hospital so injured people have nowhere to go . " The Syrian Observatory for Human Rights reports barrel bombs were dropped on the camp Sunday as clashes continued . The Palestine Liberation Organization called on international bodies to assist in the evacuation of people from the camp . `` Reports of kidnappings , beheadings and mass killings are coming out from Al - Yarmouk , which is under a brutal campaign of murder and occupation , " Palestine Liberation Organization Executive Committee Member Dr. Saeb Erekat said Saturday . Yarmouk , the largest Palestinian refugee camp in Syria , was formed in 1957 to accommodate people fleeing the Arab - Israeli conflict . `` The levels of humanity that we have seen have now descended into further levels of inhumanity , " said Chris Gunness , spokesman for the UNRWA . Yarmouk , he added , `` was always a place where human rights meant very little . We are seeing it descend further . " CNN 's Samira Said contributed to this report .

**Article 3 --- Score: 0.72**

( CNN ) They took Yarmouk by storm , a sea of masked men flooding into the streets of one the world 's most beleaguered places . Besieged and bombed by Syrian forces for more than two years , the desperate residents of this Palestinian refugee camp near Damascus awoke in early April to a new , even more terrifying reality -- ISIS militants seizing Yarmouk after defeating several militia groups operating in the area . `` They slaughtered them in the streets , " one Yarmouk resident , who asked not to be named , told CNN . `` They ( caught ) three people and killed them in the street , in front of people . The Islamic State is now in control of almost all the camp . " An estimated 18,000 refugees are now trapped inside Yarmouk , stuck between ISIS and Syrian regime forces in `` the deepest circle of hell , " in the words of U.N. Secretary - General Ban Ki - moon . Yarmouk , the largest Palestinian refugee camp in Syria , was formed in 1957 to accommodate people fleeing the Arab - Israeli conflict . The camp , which sits just 6 miles from central Damascus , has been engulfed in fighting between the Syrian government and armed groups since December 2012 . The London - based Syrian Observatory for Human Rights says ISIS and the al Qaeda - affiliated Al - Nusra Front control about 90 % of the camp . The organization also claims that the Syrian government has dropped barrel bombs on the camp in an effort to drive out armed groups . Activists and residents in Yarmouk tell CNN that as many as 5,000 people have tried to flee their homes since ISIS stormed the camp , but have no place to go . Hundreds have been injured , but the camp 's only functioning hospital was first occupied by ISIS , then targeted last week by regime shelling . As the fighting raged in Yarmouk , the director of the Jafra Foundation -- the only aid group that has been able to get into the camp -- painted a grim portrait of the conditions on the ground since ISIS arrived . `` We need medicine and access to treatment and medical facilities , " Wesam Sabaneh told CNN . `` The last hospital in Yarmouk camp was bombed yesterday , so there 's really nothing functioning . " Opinion : Save the 'miracle babies ' Even delivering clean water in Yarmouk can be a deadly task . Majed Alomari , the Jafra Foundation 's water coordinator , was killed a few days ago -- gunned down in an ISIS firefight with rival rebel groups . The head of the Palestinian League for Human Rights in Syria ( PLHR ) , who fled the camp and Syria in October 2012 , said the people of Yarmouk were in dire need of help .

Figure 14: Example highlights for Semantic method.

# D   User Study Details

## D.1   Pilots and Sample Size

Prior to conducting the actual user study, we ran pilot studies on a smaller number of participants. Using the data points collected from these studies, we conducted a Monte Carlo simulation-based power analysis to determine the effective sample size. We determined to recruit 55 participants per condition (so total of 275 = 55 × 5 conditions) for a statistical power over 0.8 with the effect size (Cohen's d) of 0.5 (orange line with circle markers in Figure 15). This effect size corresponds to 0.1 difference in the mean accuracy between the control and the treatment.

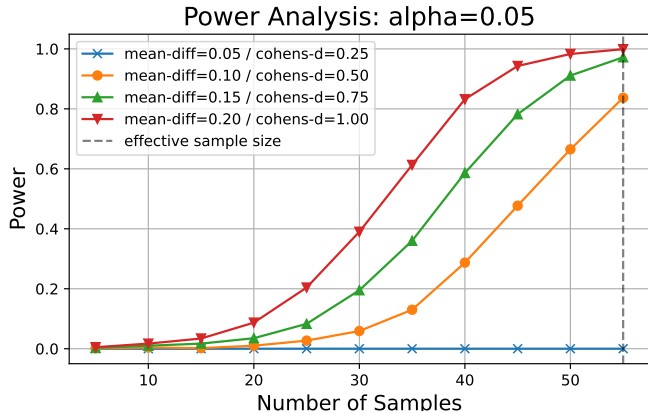

Figure 15: Power analysis for the effective sample size. We collect 55 samples per group (vertical dotted line) for a statistical power over 0.8 for the effect size (Cohen's $d$) of 0.5 (orange line with circle markers).

## D.2   Demographic Background

In Figure 16, we provide demographic background of the participants (age, ethnicity, student status, employment status) recruited for the study. 275 participants were recruited from a balanced pool of adult males and females located in the U.S. with minimum approval ratings of 90% using Prolific (`www.prolific.co`).

## D.3   Tutorial

We provide the participants with a set of instructions laying out what the highlights indicate and how one might use them for the task. The instruction is followed by two sample questions on which the participants could take unlimited time to get an understanding of what the questions look like. For the sample questions, the participants were provided the correct answers and the justification behind them as feedback.

## D.4   Payments

Base payment per participants was $3.15, determined based on the minimum hourly payment set by the platform and the median completion time of all participants, resulting in an average reward of $12.07 per hour. To encourage quicker and more accurate responses, we designed bonus payments so that each participant could earn additional $ (base payment for the question × multiplier) for each correctly answered questions, where the multiplier is determined by the response time on the question (Table 1). One could ideally earn up to ×1.5 the base payment by answering all questions correctly, all within 30 seconds. All payments (base and bonus) were processed after the data collection was complete, accounting for invalid responses.

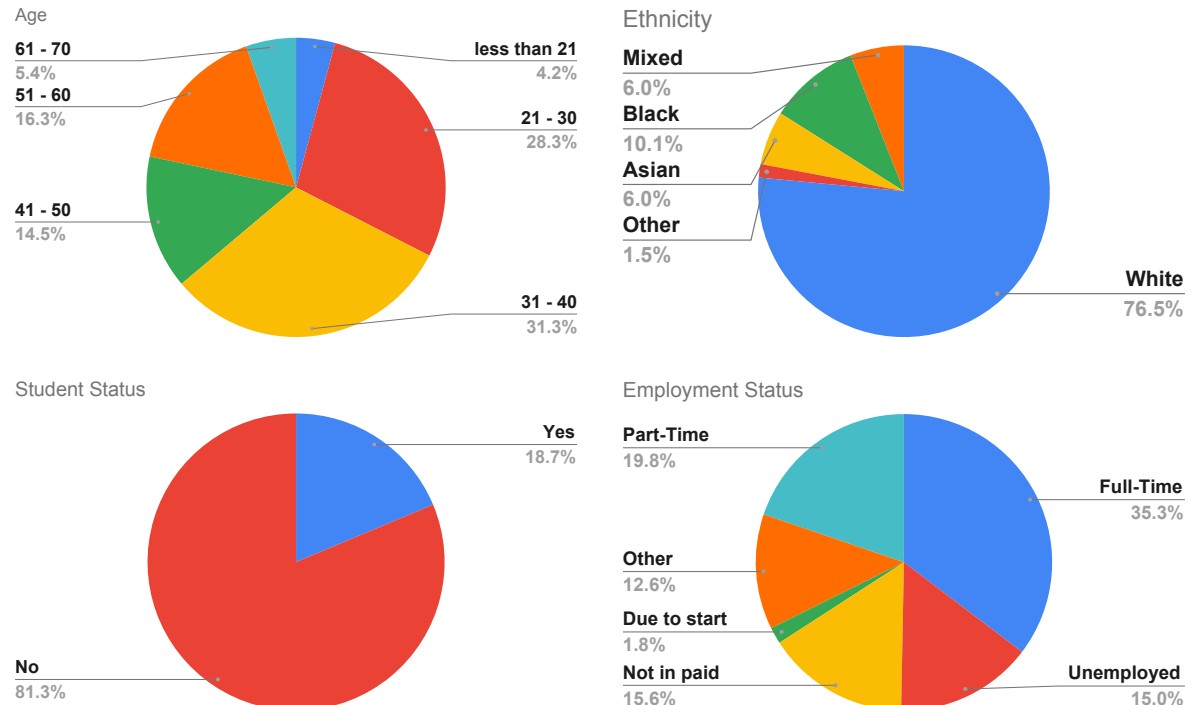

Figure 16: Demographic background of the participants (age, ethnicity, student status, and employment status).

| Response Time (seconds) | < 30 | < 60 | < 90 | < 120 | > 120 |
|---|---|---|---|---|---|
| Multiplier | x0.5 | x0.4 | x0.3 | x0.2 | x0.0 |

Table 1: Reward multiplier based on response time for correct answers. Incorrect answers have the multiplier of zero.

# E   From Summary-Article Matching to Peer Review

In this section, we describe our initial steps toward translating the summary-article matching setup to the peer review setup. We introduce a tool that uses the academic abstracts and the semantic method to assist meta-reviewers check the quality of a potential match between a submitted paper and a reviewer.

Based on the results obtained in our user study for summary-article matching, we now apply the semantic method to a more specific set of documents for peer review: academic abstracts. Abstracts contain rich and compact information about what the paper is about and are therefore widely utilized in practice. For instance, large language models use the embeddings of the abstracts to subsequently assign affinity scores to each candidate reviewer (Cohan et al., 2020).

Meta-reviewers generally "are presented with little structured information about the reviewers" (Thorn Jakobsen & Rogers, 2022): what the meta-reviewers typically see is just a list of candidate reviewers, sorted according to their affinity scores (Figure 17, left). They can then click on each reviewer's profile to see more details about the reviewer (e.g., Google scholar page, personal websites, etc.). The tool we develop, which we call R2P2 (**R**eviewer **TO P**aper for **P**eer Review), is designed to provide more structured information about each reviewer relevant to the submission. It takes in a title and an abstract of the submitted paper, and a potential reviewer's profile (Semantic Scholar profile link). It then searches for previous papers written by the reviewer and ranks each paper based on its abstract's similarity to the submitted abstract, computed

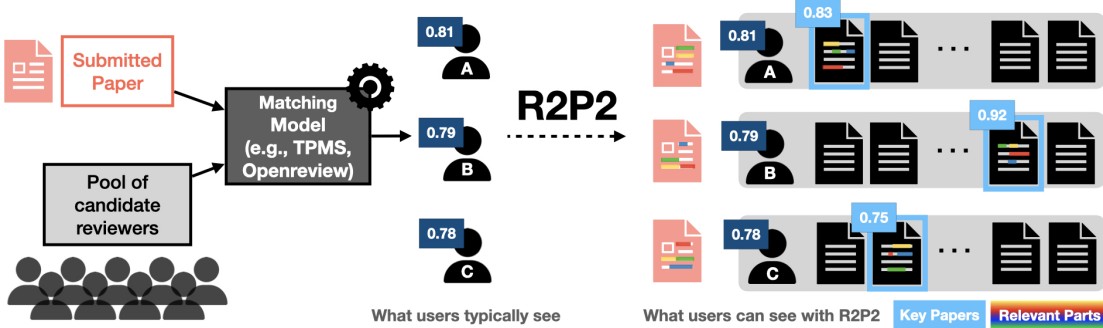

Figure 17: Overview of R2P2. While typical meta-reviewers only see the ranked list of potential reviewers and their respective affinity scores (left), R2P2 helps provide more context to the match, by searching for top relevant papers by the reviewer, as well as relevant pieces of information within those papers (right).

with embeddings from a language model of choice[8]. Once the papers are ranked and top papers are retrieved, it further uses the language model to highlight relevant sentences within the abstracts of the retrieved papers (Figure 17, right), just as we have done in the summary-article matching setup. The tool is available for trial here[9].

Unlike the summary-article matching setup where the users are simply asked to process three candidate articles and a short paragraph of summary text, this setup involves longer input texts (e.g., submitted abstracts) as well as multiple candidate texts (e.g., reviewer's previous papers) to process. This requires a change in how the highlights and relevant information are presented, as visualizing all highlights with different colors at once with all text (as we did in the summary-article matching task) is not practical. R2P2 displays a more stratified set of information as described below.

**Initial Information.** First, the users are shown a succinct characterization of what parts from the abstracts (one from the submitted paper, one from the reviewer's previous paper) make them similar. In particular, we pick the top relevant sentence pairs (based on relevance scores computed by the language model) from the submitted abstract and the abstracts of the reviewer's top papers and display them side-by-side (Figure 18). We further allow users to change the number of top papers by the reviewer and top sentence pairs to show for each paper.

**Additional Information.** If the provided information above is not sufficient, the users have a choice to explore each of the reviewer's previous papers in more detail. This interactive part gives the users more degrees of freedom to select a specific reviewer's paper, a specific sentence from the submitted abstract, and the number of relevant sentences to show from the abstract of the selected reviewer's paper (Figure 19). The red highlights indicate the relevant sentences from the selected paper's abstract, where the darkness of the color scales with the magnitude of the relevance score (the darker the higher). We highlight common phrases contained in both the submitted abstract and the reviewer's paper in blue. The users can interactively click on different parts to understand the similarities and differences between the reviewer and the submission. It is also useful to look at the list of titles on the left to quickly verify if the reviewer has published on a similar topic to that of the submission: in Figure 19, we see that two papers have high affinity scores above 0.92, all of which are about peer review.

**Preliminary Feedback.** We perform preliminary interviews with few colleagues to collect initial qualitative feedback about the tool. To make them interact with the tool in a more meaningful manner (and therefore to make them more likely to provide quality feedback), we devise a small task that emulates the meta-reviewer's job of evaluating the quality of the given paper-reviewer assignment. Specifically, they are asked to use the tool to rate the quality of 5 different paper-reviewer assignments, using a score ranging between 1 and 5 (5 being the strongest match, 1 being the weakest). The paper-reviewer assignments are sampled from a

---

[8]In our current version, we use Specter2(Cohan et al., 2020; Singh et al., 2022).

[9]https://huggingface.co/spaces/jskim/paper-matching

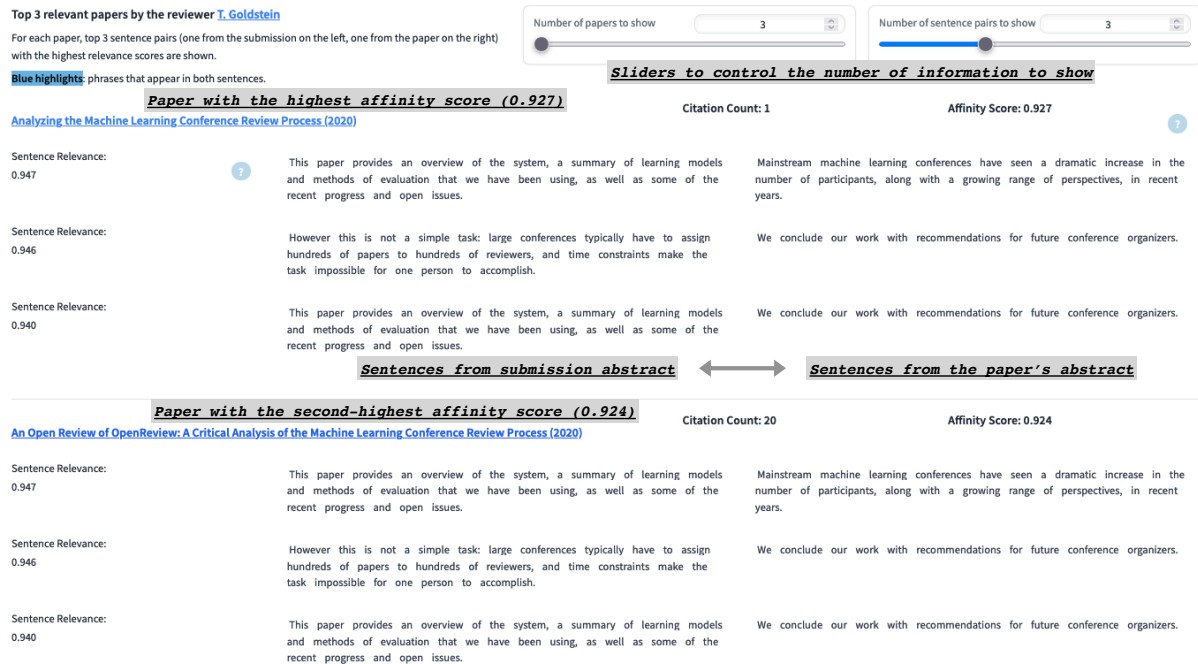

Figure 18: R2P2 first provides relevant sentence pairs from the submission abstract (left) and the top papers by the reviewer (right). Example here is for the TPMS paper (Charlin & Zemel, 2013) as the submission and Tom Goldstein (`https://www.semanticscholar.org/author/T.-Goldstein/1962083`) as the reviewer. From the presented information, it is easy to catch that both the reviewer and the submission have a common interest in conference review process.

gold-standard dataset, which consists of "self-reported expertise scores provided by researchers who evaluated their expertise in reviewing papers they have read previously" (Stelmakh et al., 2023).

Generally, the feedback was positive, with suggestions for improving the UI and the algorithm. One common feature that the participants found particularly useful was the title of top relevant papers from the reviewer's profile. Part of the common workflow was to go through the list of these titles to deduce how the reviewer may be relevant. It was also pointed out that skimming through the reviewer's publication list manually without such information would make it easy to miss few critical papers that are highly relevant. Additional information with a more interactive UI was also perceived as helpful with the highlights capturing important details for the task.

Sentence-by-sentence comparison presented for the initial information was generally useful, but the participants suggested it could be improved in terms of selecting the sentences to display. There were cases where the selected sentence pairs were introductory sentences of respective abstracts, which contain less specific information about the assignment. While they were useful in checking for a general alignment of expertise, more detail was needed to ensure the quality of the match, which was more effectively captured in the interactive part of the tool. One suggested a possibility of improving this by leveraging a structural nature of academics abstracts (e.g., first sentence of the abstract is almost always about the general field of study; last few sentences mostly discuss specific details that are unique to the paper) to diversify the selection of presented information.

Next steps would include addressing the feedback to improve the tool, and running a scaled-up controlled study with domain experts to rigorously validate the usefulness of the tool. The gold-standard dataset (Stelmakh et al., 2023) contains multiple paper-reviewer assignments that can be utilized to design such a study. However, one needs to modify the task format so that it accounts for calibration issues, as the scores given in the

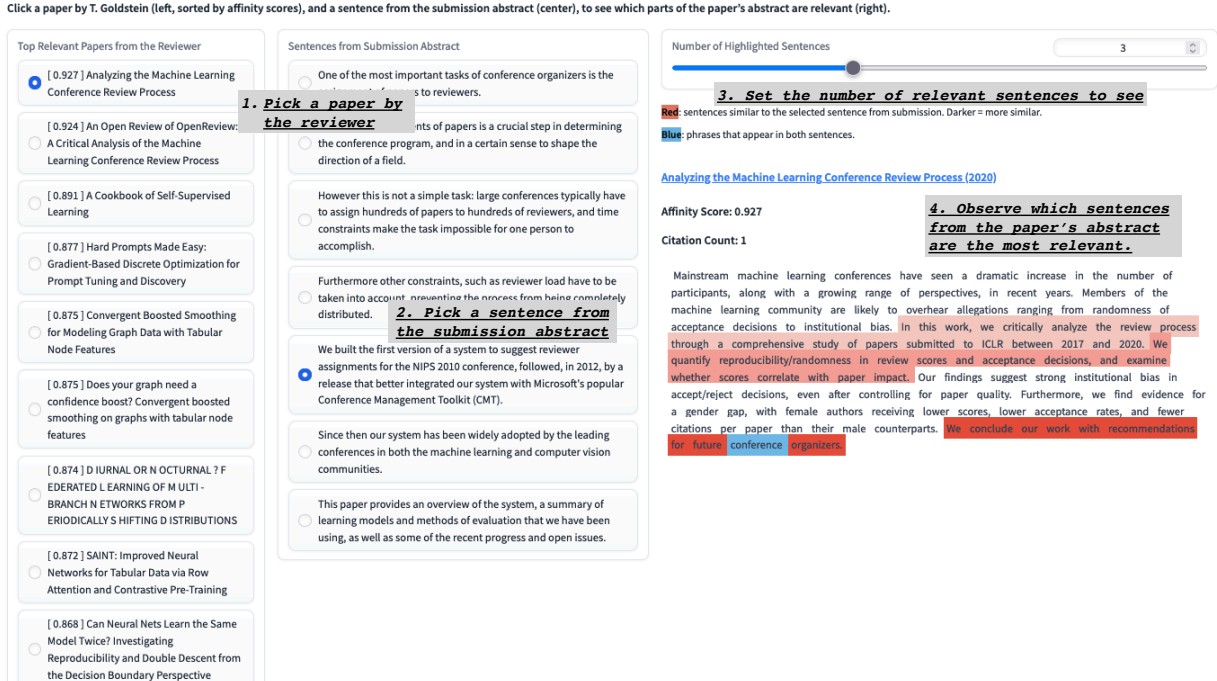

Figure 19: R2P2 then allows users to interactively explore more details for each of the reviewer's paper. One can select a particular paper, select a particular sentence from the submission abstract, to see which sentences of the paper's abstract are the most relevant to the selected sentence.

datasets will be different from scores given by the participants. Additionally, setting up the study would require more care due to the high cost of recruiting domain experts in scale.

