# OpenReview forum: "Assisting Human Decisions in Document Matching"
_TMLR — Accepted by TMLR_

### Review · Reviewer_k9gw · 2023-03-10

**Summary Of Contributions:**

The study finds a surprising result that providing users with black box explanations reduces their accuracy in matching tasks, contrary to conventional intuition. Additionally, the author finds that user's perception of the assistance and its objective utility is misaligned.

However, I have some concerns. To ensure the generalizability of the conclusions, it would be worthwhile to consider the performance of SHAP and BERTSum, two general methods, and two task-specific methods on other proxy tasks, which may yield different results. Furthermore, the details of SHAP and BERTSum methods were not discussed, and it is unclear how the model's effectiveness is guaranteed.

The author mentioned that SHAP's performance varies in different downstream tasks, so other stable SOTA methods should be added for comparison. In addition, the author highlighted relevant information in the candidate article without considering users' acceptance of this method itself, so it may be beneficial to add other forms of assistance.

**Audience:**

No

**Broader Impact Concerns:**

The paper may have limited impacts in the machine learning community.

**Claims And Evidence:**

Yes

**Requested Changes:**

See the above comments.

**Strengths And Weaknesses:**

See the above comments.

---

> ### Author Response · Authors · 2023-05-24
> **Thank you for the thoughtful feedback. We address individual points separately.**
>
> Thank you for the thoughtful feedback. We address individual points separately below:
>
> >  To ensure the generalizability of the conclusions, it would be worthwhile to consider the performance of SHAP and BERTSum, two general methods, and two task-specific methods on other proxy tasks, which may yield different results.
>
> The summary-article matching task was designed to reflect many of the nuances of the general document matching tasks (Section 3.1). While designing another proxy task that resembles the general document matching task could provide further insights, given that this new proxy task is likely to be quite similar to the summary-article matching task, we find it difficult to justify the high cost of running a separate user study on it. We note that SHAP in particular has already been applied to multiple different tasks with negative results as mentioned in Section 2.
>
> Rather than designing additional proxy tasks, we have taken preliminary steps to translate our proposed methods into a tool for more practical tasks like peer review (Appendix E of the revision). We take our proposed method and integrate it into a tool that can be readily used by the meta-reviewers for the actual task of assessing the quality of the match suggested. Specifically, given a submission abstract and a profile of a potential reviewer, the tool first searches for the relevant papers from the reviewer’s publication history, and then highlights relevant parts within. The difference in the amount of text compared to the summary-article matching task (originally mentioned in Section 5) was also taken into account when designing the UI of the tool. The preliminary qualitative feedback from a small number of colleagues who tried the tool was positive. Nevertheless, it remains a future work to rigorously validate the usefulness of the tool through a pre-registered controlled user study with domain experts.
>
> > Furthermore, the details of SHAP and BERTSum methods were not discussed, and it is unclear how the model's effectiveness is guaranteed.
>
> We added more details of each method in Appendix B in the revision. We include links to the source code and libraries used for the implementation of these methods as well as their configurations, and how these methods were applied to the text to generate outputs.
>
> > Other stable SOTA methods should be added for comparison.
>
> In our original submission, we considered different canonical methods for black-box model explanations and explained why SHAP was chosen for the user study (Appendix A). Unfortunately, at the moment, there exist no widely-accepted methods that are more stable and clearly better than others. Accordingly, SHAP was chosen as a representative example, given its active use in a broad range of applications like biology, physics, chemistry, and finance (described in Appendix A). We will release the link to the data and code for the proxy task with the camera-ready version of the paper, which can be later utilized to test new methods of interest.
>
> > The author highlighted relevant information in the candidate article without considering users' acceptance of this method itself, so it may be beneficial to add other forms of assistance.
>
> In general, studying questions of human adherence or reliance on assistive information and decision support is an ongoing area of research. We tried to mitigate the issues by onboarding the participants with detailed instructions, tutorials, and sample questions. As mentioned in Appendix D.3, before the user study, we provided instructions on what the highlights are, and how one should go about using them. Additionally, we provided the participants with two practice sample questions prior to the beginning of the study to have them familiarize themselves with the interface and to minimize potential confusion.

---

### Review · Reviewer_pwvc · 2023-05-08

**Summary Of Contributions:**

The paper considers the problem of document matching. In the designed task, participants are given a summary and have to choose one between three possible texts that could have generated the summary.

**Audience:**

Yes

**Broader Impact Concerns:**

-possible application to hiring procedure.

**Claims And Evidence:**

No

**Requested Changes:**

-Analyze more in depth the easy task as well
-consider enriching the experimental setup, including scores that match the adopted assistive technique

**Strengths And Weaknesses:**

Strengths:
the research question faced by the authors is interesting since it is unclear how to integrate humans and machine learning techniques in practical scenarios.

Weaknesses:
First, the authors present the considered document matching task as a proxy task for other decision-making processes such as academic peer review or HR. It is not clear how well this proxy task is aligned to these other tasks, and how well the proposed techniques can be applied to these other domains (e.g. curricula are way different from the documents considered in the paper).

An important point for my following comments is that the three texts are chosen using a language model: 1) the true text associated to the summary and 2) the two highest scoring, incorrect texts (i.e. texts that would be associated with a different summary).

The experimental setup considers two settings: easy and hard.

In the easy setting, the correct text is always associated with the highest score, while in the hard setting this is almost never the case.

Authors do not spend much time underlining this difference. They even do not consider the easy setting in the considered research hypotheses. However, I think it is of key importance.
In the easy setting, the model used to select the sentences is correct.
Thus, methods that explain the model decision work well. In fact, from Figure 6 the performances of the different methods are more or less in line.

In the hard task, the model is mainly wrong. Thus, the explanation methods considered are "justifications" for the (wrong) model scores. It doesn't surprise me that those explanations are confusing for a user. Moreover, the scores are not aligned with the assistive information.
In fact, the two ad-hoc methods proposed by the authors and BERTsum can be seen as building blocks of an alternative model.
For BERTsum, one can compute a score between the generated summary and the ground-truth one.
The proposed models would perform hard (Co-occurrence) or soft (Semantic) matching between the summary and the texts. One can then figure out a way to compute an overall similarity score from those measures. I would expect this model to perform better than the adopted DistilBART.

The main consideration is that the considered methods do not reflect the score shown to the user, but would reflect another induced score that is not shown. How would users behave when the score is actually aligned with the assistive information?


Minor comments:

Page 3: please add more details about the ROUGE metric.

---

> ### Author Response · Authors · 2023-05-24
> **Thank you for the constructive feedback. We address individual points separately.**
>
> Thank you for the constructive feedback. We address individual points separately below:
>
> > It is not clear how well this proxy task is aligned to these other tasks, and how well the proposed techniques can be applied to these other domains.
>
> The proxy task (and associated evaluation) was designed to reflect many of the constraints/parameters that occur in real-world use cases that we motivate the task with as outlined throughout Section 3. The proxy task’s purpose, as outlined at the beginning of Section 3, is to make the task “more amenable for crowdsourcing experiments at scale”, so that we can efficiently screen a potential set of methods that may be useful in practical (and more costly) settings like peer review.
>
> In Appendix E of the revision, we present our preliminary attempt at extending to a more practical setting of peer review. We take our proposed method and integrate it into a tool that can be readily used by the meta-reviewers for the actual task of assessing the quality of the match suggested. Specifically, given a submission abstract and a profile of a potential reviewer, the tool first searches for the relevant papers from the reviewer’s publication history, and then highlights relevant parts within. Preliminary qualitative feedback from a small number of colleagues who tried the tool was positive. Nevertheless, it remains a future work to rigorously validate the usefulness of the tool through a pre-registered controlled user study with domain experts.
>
> > In the easy setting, the model used to select the sentences is correct. Thus, methods that explain the model decision work well. In the hard task, the model is mainly wrong. Thus, the explanation methods considered are "justifications" for the (wrong) model scores. It doesn't surprise me that those explanations are confusing for a user.
>
> We agree with the reviewer’s reasoning for why the model-based method (e.g., SHAP) works better on the easy task compared to the hard task and included a similar discussion in Section 5 of the revision. However, we clarify that SHAP, along with all other methods discussed, is still not “useful” for the easy questions because they lead to comparable accuracy with the Control group while not reducing the amount of time spent. This contrasts with the results we observe in the hard task, where there are clear gains in accuracy and time from methods other than SHAP compared to the Control group.
>
> > Analyze more in depth the easy task as well. They even do not consider the easy setting in the considered research hypotheses.
>
> While we do not include statistical tests for the easy task, we revised the submission to discuss the related results more in Section 4.3.1. We added that the diverging results in the easy and hard settings for the same methods demonstrate different scenarios for which specific methods are more effective than others. For the easy task, relying on the model can be effective, while for the hard task, additional assistance from the Semantic method can be particularly effective.
>
> We do not include statistical tests for the easy task because they were not part of the pre-registration of the experiment. This was for two reasons. First, if the tests for the easy task were to be included, the effective sample size would have become much larger to draw reliable conclusions about the methods (taking multiple testing into account), increasing the cost of the experiment significantly. Second, we were more interested in the methods’ ability to assist with the hard task as such a situation is more likely to call for additional assistive information. Hence, we prioritized testing the hard task. We added a discussion in Section 4.2 explaining the above points.
>
> >  The considered methods do not reflect the score shown to the user, but would reflect another induced score that is not shown. Consider enriching the experimental setup, including scores that match the adopted assistive technique.
>
> The primary focus of the work is to test if assistive information is useful for the document matching task, which does not require the method to strictly reflect the score shown to the user. We observed that even when the assistive information is not directly tied to the affinity scores shown, the proposed methods already improve decision accuracy. We note the reviewer’s suggestion as an interesting direction for future work in Section 5 of the revision.
>
> > Broader impact concerns in possible application to hiring procedure.
>
> We added a broader impact statement regarding this in the revision.
>
> > Page 3: please add more details about the ROUGE metric.
>
> We added more details about the ROUGE metric on page 3: ROUGE is a statistic used to measure the degree of the overlap of tokens in strings, which can be computed using different software libraries. We also include more details on what library was used and how the metric was computed for our use case in Appendix B.

---

### Review · Reviewer_A385 · 2023-05-12

**Summary Of Contributions:**

Paper deals with the combination of expertise with predictions from machine learning models to identify relevant matches in applications such ad peer review or hiring using assistive information about the model outputs to facilitate their decisions.
Paper, propose a method that allows us to evaluate which kinds of assistive information improve decision makers’ performance (in terms of accuracy and time).
Through and experimental study with 271 participants authors
- show that black-box model explanations reduces users’ accuracy on the matching task, contrary to the commonly-held
belief that they can be helpful by allowing better understanding of the model.
- using custom methods that are designed to closely attend to some task-specific desiderata are found to be effective in improving user performance
- show that the users’ perceived utility of assistive information is misaligned with their objective utility

**Audience:**

Yes

**Broader Impact Concerns:**

I do not see any ethical concerns.

**Claims And Evidence:**

Yes

**Requested Changes:**

I think that authors should include in the appendix more details about he setting ofthe algorithms to obtain the results (e.g., hyperparameters search, etc.).

**Strengths And Weaknesses:**

I think that the paper is concise, clear, straight to the point.
The only limitation here is that there is quite limited details about the experimental setup (e.g., setting of the algorithms, etc.)

---

> ### Author Response · Authors · 2023-05-24
> **Thank you for the thoughtful feedback. We added more details in Appendix B.**
>
> Thank you for the thoughtful feedback. We have added additional details about the experimental setup of each method in Appendix B of the revision. We used default hyperparameter settings for the methods in their native implementations. We include links to the source code and libraries used for the implementation and how these methods are applied to the text to generate outputs.

---

### Decision · Action_Editors · 2023-06-18

**Recommendation:** Accept as is

**Comment:**

The paper addresses a relevant and timely issue concerning the application of machine learning techniques to support human activities for a specific class of applications involving document matching. Although the study focuses on a specific task, the findings of the described study are clearly described, supported by a  good amount of experimental evidence, and already by themselves constitute an interesting contribution to the area. Some presentation issues raised by the reviewers have been properly addressed by the revision of the paper.

**Audience:**

Although the paper does not cover a core issue in machine learning, it covers a topic that is very relevant when deploying NLP machine learning solutions (specifically, document matching) in practical applications. I guess there will be several individuals in TMLR's audience  interested in the findings of the paper.

**Claims And Evidence:**

The claims of the paper are convincing and seems to be accurate. The claims are also supported by a good amount of experimental evidence.